# PAMELA: Probabilistic Amplification Module for Lightweight AI on LEO Satellites

## Abstract

Onboard AI inference in low Earth orbit (LEO) satellites enables rapid analysis of Earth observation data for time-critical applications such as wildfire monitoring. Yet, two key challenges remain: (i) critical fire signals are extremely sparse, often confined to only a few pixels, and (ii) onboard models must be highly compact due to the stringent hardware constraints of LEO platforms. As a result, small models may report deceptively high overall accuracy by overfitting to background regions, while suffering from poor precision and recall on fire pixels. To overcome these limitations, we introduce PAMELA, a lightweight and modular amplification framework tailored to enhance the performance of onboard wildfire detection models under resource-constrained conditions. PAMELA employs probabilistic modeling to selectively amplify informative channels and pixels, allowing compact models to better capture sparse but mission-critical signals within complex satellite imagery. Experimental results demonstrate that PAMELA consistently improves detection quality, delivering $1.2 \sim 2\times$ higher F1 scores compared to compressed baselines while simultaneously reducing model size by over 90%. To the best of our knowledge, PAMELA is the first framework explicitly designed to enable reliable onboard wildfire detection in LEO satellites.

## 1 Introduction

The widespread deployment of low Earth orbit (LEO) satellites has enabled continuous, high-frequency Earth observation for diverse applications, including environmental monitoring Zhang et al. (2025); Smith et al. (2025); Rahman et al. (2023), disaster assessment Gomez et al. (2024); Clark et al. (2024); Anderson et al. (2023), and land use analysis Brown et al. (2025); Nguyen et al. (2024). Among these, wildfire detection is one of the most time-critical applications. Current operational pipelines typically transmit raw images to ground stations for subsequent processing. This strategy, however, introduces significant delays due to limited downlink bandwidth and ground-based processing bottlenecks, undermining timely response in fast-evolving wildfire scenarios Jin et al. (2022); Zhou et al. (2023). To reduce this latency, it is essential to explore the feasibility of reliable onboard wildfire detection, where inference is performed directly on the satellite before transmission.

Onboard wildfire detection, however, faces two fundamental challenges. The first is hardware constraints—restricted memory, computation, and energy budgets—which necessitate the use of highly compact neural models Smith et al. (2025); Rahman et al. (2023). Such constraints make it difficult to execute vision tasks on dense, high-resolution, and multi-spectral satellite imagery. The second challenge lies in the characteristics of wildfire signals: they are spatially sparse, irregular, and dynamic, often occupying less than 1% of pixels in a scene Zhang et al. (2025); Mohammed et al. (2025). Under these conditions, lightweight models tend to overfit to the dominant background class, producing overconfident false negatives and poor recall. Consequently, models may achieve deceptively high overall accuracy while systematically overlooking task-critical fire patterns Martinez et al. (2023); Clark et al. (2024).

Existing approaches for wildfire detection from satellite imagery have primarily relied on ground-based processing. In recent years, deep learning models—including U-Net, SegNet, FCN-based architectures, and vision transformers—have been widely adopted to achieve pixel-level segmentation of fire regions Mohammed et al. (2025); Clark et al. (2024); Khan et al. (2024). While these

methods deliver state-of-the-art accuracy in ground-based environments, they are typically too large and computationally demanding to be deployed directly on resource-constrained satellites. To address this, model compression techniques such as pruning, quantization, and knowledge distillation have been explored Han et al. (2016); Cheng et al. (2018). However, when models are compressed to the extremely small sizes required for satellite deployment, accuracy often drops sharply. In particular, quantization can significantly degrade sensitivity by sacrificing numerical precision, leading to poor recall on the sparse fire pixels that are most critical for timely detection. This limitation highlights the need for novel onboard techniques that can preserve high recall on task-critical wildfire signals while remaining efficient under extreme resource constraints.

To address this challenge, we introduce the **P**robabilistic **A**mplification **M**odul**E** for **L**ightweight **AI** on LEO Satellites (PAMELA), a modular enhancement block designed to boost the performance of compressed wildfire detection models. To the best of our knowledge, **PAMELA is the first framework explicitly tailored for reliable onboard wildfire detection in LEO satellites.** Rather than acting as a standalone model, PAMELA integrates into existing small backbones—such as those pruned for satellite deployment—and is trained jointly in an end-to-end fashion. It enhances inference by estimating a pixel-wise likelihood map that reflects the statistical similarity of each pixel to expected fire signals across spectral bands. This map reweights the input image, enabling the compressed model to allocate its limited capacity toward informative regions while suppressing background noise.

We validate PAMELA on wildfire detection tasks across a range of compressed backbones. PAMELA consistently improves F1 score by $1.2\times$–$2\times$ with less than 1% additional parameters, often matching or surpassing larger uncompressed baselines. These results confirm PAMELA's utility as a universal, resource-efficient enhancement for onboard AI. Our contributions are summarized as follows:

- We introduce PAMELA, a lightweight probabilistic amplification module specifically designed to enhance the performance of small models for onboard wildfire detection in LEO satellites. To our knowledge, this is the first work to address the challenge of sparse-signal amplification in wildfire imagery under onboard computational constraints. PAMELA fills this gap by enabling real-time prioritization of fire-relevant regions through adaptive pixel-wise reweighting.

- PAMELA not only improves feature focus but also enhances interpretability. The pixel-wise reweighting process in PAMELA provides a direct measure of each pixel's importance, allowing us to visualize the model's focus during inference. This property offers transparency in decision-making.

- PAMELA is a model-agnostic module designed for integration into any backbone architecture. While no powerful onboard wildfire detection models currently exist, PAMELA is forward-compatible and can be integrated into future, more capable AI architectures. Its lightweight design and adaptability allow it to enhance performance without requiring architecture-specific modifications.

## 2 DESIGN

### 2.1 OVERVIEW OF PAMELA

The PAMELA framework is a lightweight and modular probabilistic amplification method designed to enhance the performance of small models on LEO satellites by adaptively reweighting pixels in input images. As illustrated in Figure 1(a), PAMELA begins with a multi-channel input image, which is first processed by a **Channel Filter** to identify and retain the most informative spectral bands. Each selected channel is then independently passed through a **Probabilistic Amplifier** to generate channel-wise pixel-level probability maps. These maps, together with the filtered input, are fused via a Multi-variate PA to produce a unified probabilistic representation. Finally, a **Reweighting Block** transforms this probability map into a pixel-wise reweighting matrix, which is applied to the original input. The reweighted image is subsequently fed into a lightweight backbone model, as shown in Figure 1(b). Importantly, PAMELA is not applied on top of a pre-trained or frozen model; instead, all components—including PAMELA and the base model—are trained jointly in an end-to-end fashion. This co-optimization ensures that the amplification process evolves in sync with the

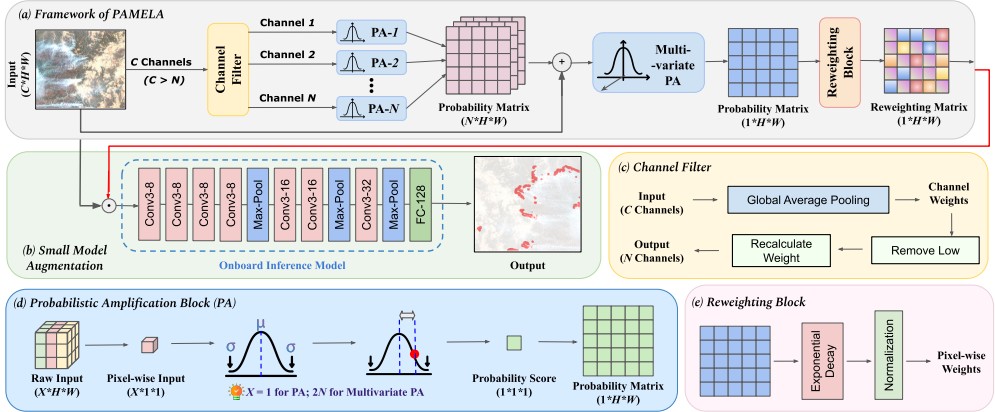

Figure 1: **Workflow of PAMELA Framework.** (a) PAMELA begins with an input containing multiple channels, processed by the Channel Filter to retain important ones. Each selected channel is passed through a Probabilistic Amplification Block (PA) to generate a probability matrix. These matrices and the raw input are combined in a Multi-variate PA, and the Reweighting Block converts the result into a reweighting matrix. (b) The reweighted image is processed by a lightweight onboard inference model for final output. (c) The Probabilistic Amplification Block (PA) calculates pixel-wise probabilities using a learned probabilistic model, with $\mu$ and $\sigma$ learned during training. (d) The Channel Filter uses Global Average Pooling to calculate channel importance, recalculating weights and removing channels with low importance. (e) The Reweighting Block converts the probability matrix into a pixel-wise reweighting matrix through exponential decay and normalization.

model's learning dynamics, allowing it to effectively highlight task-relevant regions that improve downstream inference.

## 2.2 CHANNEL FILTER

As shown in Figure 1(c), the Channel Filter is the first component of the PAMELA framework, designed to automatically identify and retain only the most informative channels from the multi-channel input image $\mathbf{X} \in \mathbb{R}^{C \times H \times W}$, where $C$ is the number of channels, and $H \times W$ denotes spatial resolution. To measure channel relevance, we first apply Global Average Pooling (GAP) across the spatial domain Lin et al. (2014); He et al. (2016); Howard et al. (2017); Hu et al. (2018), generating an initial descriptor vector $\mathbf{g} \in \mathbb{R}^C$, where each element encodes the average activation strength of a given channel:

$$g_c = \frac{1}{H \times W} \sum_{h=1}^{H} \sum_{w=1}^{W} X_{c,h,w}, \quad \forall c \in \{1, 2, \ldots, C\} \tag{1}$$

However, raw average intensity is not always aligned with semantic usefulness. To better distinguish truly task-relevant channels, we introduce a nonlinear gating function that maps channel intensities to sparse selection weights:

$$\alpha_c = \text{ReLU}\left(\tanh\left(w \cdot \hat{g}_c + b\right)\right) \tag{2}$$

where $\hat{g}_c$ denotes the min-max normalized version of $g_c$, and $w, b \in \mathbb{R}$ are learnable scalar parameters shared across all channels. These scale and shift factors are optimized jointly with the rest of the network via backpropagation. The input is then reweighted by these per-channel weights:

$$X_{c,h,w}^{\text{weighted}} = \alpha_c \cdot X_{c,h,w}, \quad \forall c \in \{1, 2, \ldots, C\} \tag{3}$$

Channels with $\alpha_c = 0$ are effectively dropped, while informative channels are preserved and emphasized. The resulting filtered tensor $\mathbf{X}^{\text{filtered}} \in \mathbb{R}^{\tilde{C} \times H \times W}$ includes only active channels $\tilde{C} \leq C$, allowing the rest of the pipeline to operate on a reduced but information-rich representation.

This lightweight and fully differentiable filtering process allows PAMELA to adaptively select spectral bands based on data-driven relevance rather than hard-coded heuristics.

## 2.3 PROBABILISTIC AMPLIFIER

As shown in Figure 1(d), the Probabilistic Amplifier (PA) is the core component of the PAMELA framework, designed to enhance the focus of the model on task-critical regions within the image. After the Channel Filter has retained the most informative channels, each of these selected channels is independently processed by a PA block. The primary objective of PA is to reweight each pixel in the selected channels based on its likelihood of matching task-relevant characteristics. Then the model could prioritize critical regions while suppressing background noise. Specifically, for each pixel in the selected channels $\mathbf{X}^{\text{filtered}} \in \mathbb{R}^{N \times H \times W}$, the PA block computes a pixel-wise probability matrix $\mathbf{P}_n$ for each channel $n \in \{1, 2, \ldots, N\}$. This is achieved using a learned probabilistic model, which estimates the likelihood of each pixel value being task-relevant:

$$P_{n,h,w} = \exp\left(-\frac{(X_{n,h,w} - \mu_n)^2}{2\sigma_n^2}\right) \tag{4}$$

Here, $X_{n,h,w}$ denotes the pixel value at position $(h, w)$ in channel $n$. $\mu_n$ and $\sigma_n^2$ are the mean and variance parameters, which are learnable and optimized during training. This formulation assigns higher probability values to pixels whose intensity is closer to the expected value $\mu_n$, effectively amplifying pixels that are more likely to be task-critical. After each channel is processed by its individual PA block, the resulting probability matrices $\mathbf{P}_1, \mathbf{P}_2, \ldots, \mathbf{P}_N$ are combined with the input $\mathbf{X}^{\text{filtered}}$ to form a multi-channel tensor:

$$\mathbf{X}^{\text{multi}} = \text{Concat}(\mathbf{X}^{\text{filtered}}, \mathbf{P}_1, \mathbf{P}_2, \ldots, \mathbf{P}_N) \tag{5}$$

This combined tensor $\mathbf{X}^{\text{multi}} \in \mathbb{R}^{(C+N) \times H \times W}$ is then passed through a Multi-Variate PA block, which directly operates on this multi-channel input to produce a unified probability matrix:

$$P_{h,w}^{\text{multi}} = \exp\left(-\frac{1}{2}\sum_{n=1}^{N}\frac{(X_{n,h,w} - \mu_{n,h,w})^2}{\sigma_{n,h,w}^2}\right) \tag{6}$$

Here, $\mu_{n,h,w}$ and $\sigma_{n,h,w}^2$ are learnable parameters for the Multi-Variate PA block, allowing it to capture inter-channel relationships. This Multi-Variate PA can be interpreted as modeling the joint probability of pixels being task-relevant across all channels, making it more expressive than a single-channel PA. To ensure numerical stability and maintain the probabilistic nature of the outputs, the final probability matrix $P^{\text{multi}}$ is normalized using a softmax operation:

$$\hat{P}_{h,w}^{\text{multi}} = \frac{P_{h,w}^{\text{multi}}}{\sum_{h=1}^{H}\sum_{w=1}^{W}P_{h,w}^{\text{multi}}} \tag{7}$$

## 2.4 REWEIGHTING BLOCK

As shown in Figure 1(e), the Reweighting Block converts the unified probability matrix $\hat{P}^{\text{multi}} \in \mathbb{R}^{H \times W}$ into a pixel-wise reweighting matrix $\mathbf{W}$ using a localized contrast-aware normalization. Instead of applying global softmax, which can dilute sparse activations, we propose a formulation that emphasizes local saliency within spatial neighborhoods. Specifically, for each pixel location $(h, w)$, we compute:

$$\mu_{h,w} = \frac{1}{|\mathcal{N}_{h,w}|}\sum_{(i,j)\in\mathcal{N}_{h,w}}\hat{P}_{i,j}^{\text{multi}} \tag{8}$$

$$W_{h,w} = \sigma\left(\hat{P}_{h,w}^{\text{multi}} - \mu_{h,w}\right) \tag{9}$$

Here, $\mathcal{N}_{h,w}$ denotes a fixed-size square neighborhood (e.g., 5×5 patch) centered at $(h, w)$, and $\sigma(\cdot)$ is the sigmoid function to ensure normalized weights in $[0, 1]$. This contrast-based formulation ensures that a pixel receives a high weight only if its predicted relevance exceeds its local average, thereby suppressing smooth or ambiguous regions while preserving compact activations.

The resulting matrix $\mathbf{W} \in \mathbb{R}^{H \times W}$ is then applied to the original input $\mathbf{X} \in \mathbb{R}^{C \times H \times W}$ via channel-wise multiplication:

$$\mathbf{X}_{c,h,w}^{\text{reweighted}} = X_{c,h,w} \cdot W_{h,w}, \quad \forall c \in \{1, 2, \ldots, C\} \tag{10}$$

This lightweight, differentiable transformation dynamically adapts to spatial context, amplifying sharp localized changes, such as wildfire ignition points.

## 3 EXPERIMENTS AND RESULTS

### 3.1 EXPERIMENTAL SETUP

We conduct a controlled evaluation across a diverse set of compressed backbone models to simulate the constraints of onboard inference in LEO satellites. For each architecture, we first finetune the full-capacity (uncompressed) model on the target wildfire detection task to establish a strong baseline. We then apply three state-of-the-art compression strategies to derive lightweight variants:

- **Channel reduction.** We systematically reduced the width of convolutional layers by factors of $25\%$, $50\%$, and $75\%$, thereby producing feature maps that are substantially narrower while keeping the encoder–decoder depth intact. In our experiments, reducing channels by $50\%$ lowered the FLOPs by $65\%$ on average and shrank parameter counts by nearly $3\times$, while still preserving the overall architectural structure.
- **Pruning.** We applied structured filter pruning based on weight magnitude, progressively removing $30\%$, $50\%$, and up to $80\%$ of convolutional filters. After pruning, the networks typically retained only $20$–$40\%$ of the original active parameters, resulting in $2$–$4\times$ FLOP reductions.
- **Knowledge distillation.** We trained compact student models ($1/8$–$1/10$ the size of the teacher) to mimic the logits of their original counterparts using soft targets. The student networks achieved up to $8\times$ lower computational cost.

Each compression strategy produces a model serving as the baseline backbone for integrating PAMELA, which is trained jointly with the backbone in a fully end-to-end manner. Importantly, the compressed models preserve the original encoder–decoder structure (e.g., downsampling blocks, bottleneck, and upsampling path), but with narrower channels, pruned filters, and distilled intermediate representations. This ensures a fair comparison where PAMELA operates under realistic onboard conditions while maintaining architectural compatibility across different backbones.

**Metrics.**  For evaluation, we report both detection quality and efficiency: pixel-level IoU, precision, recall, and F1 score to capture accuracy on sparse fire regions, alongside FLOPs and parameter counts to reflect computational and memory requirements under realistic onboard constraints.

**Hardware configuration.**  To reflect the memory and compute constraints of real LEO platforms, all experiments are conducted on hardware comparable to that used in prior satellite-AI simulation studies Lu et al. (2023); Zheng et al. (2022), specifically a Raspberry Pi 4B (4 GB RAM, quad-core ARM Cortex-A72 CPU) configured as an onboard processor proxy. This setup has been widely adopted in recent literature to emulate satellite-grade resource limitations during model development and validation.

**Dataset.**  In this experiment, we use the Sen2Fire dataset Xu et al. (2024), a challenging benchmark constructed from Sentinel-2 multispectral imagery and Sentinel-5P aerosol products. Sen2Fire focuses on four major wildfire events from the 2019–2020 Australian bushfire season in New South Wales, Australia. It contains 2466 image patches (512×512 pixels), each with 13 channels—12 multispectral bands from Sentinel-2 and one aerosol index from Sentinel-5P—and includes pixel-level wildfire labels derived from MODIS fire products. For our experiments, we randomly divide the dataset into 70% training, 15% validation, and 15% testing, ensuring that image patches from the same geographic region are not split across different sets to avoid spatial leakage.

**Baseline models.**  We evaluate PAMELA by integrating it into four widely used semantic segmentation models in remote sensing: **FCN** Shelhamer et al. (2016), **SegNet** Badrinarayanan et al. (2017), **U-Net** Ronneberger et al. (2015), and the transformer-based **SegFormer** Xie et al. (2021). These architectures form the foundation for many modern detection systems across tasks such as land cover mapping, disaster monitoring, and environmental change analysis.

FCN performs end-to-end segmentation by replacing fully connected layers with convolutions, enabling dense predictions Li et al. (2023); Zeng et al. (2023). SegNet uses an encoder–decoder structure with index-based upsampling to reduce memory usage Badrinarayanan et al. (2017), while U-Net enhances spatial resolution through skip connections for fine-grained segmentation Ramos et al.

(2025). SegFormer employs a hierarchical transformer encoder combined with lightweight MLP decoders, enabling efficient long-range dependency modeling while maintaining compactness Xie et al. (2021). However, when compressed for LEO deployment, all four architectures suffer from reduced spatial precision, contextual modeling, and hierarchical feature integration—highlighting the challenge of maintaining performance under strict resource constraints Yin et al. (2024).

## 3.2 RESULTS

The results in Table 1 and Figure 2 consistently demonstrate that while conventional compression strategies (channel reduction, pruning, and distillation) achieve substantial efficiency gains in terms of computation and model size, they often do so at the cost of significant drops in segmentation accuracy, F1 score, and recall. For example, across all four architectures, compressed models without enhancement lose between 5–15 points in accuracy and F1 compared to their original counterparts. In contrast, when augmented with our proposed PAMELA, the compressed models recover most of this lost accuracy and, in many cases, even match or slightly exceed the original models' detection performance. This effect is particularly pronounced in low-capacity backbones such as SegNet and U-Net, where vanilla compression causes drastic degradation (e.g., SegNet channel-reduced F1 dropping to 9.5%), but PAMELA lifts performance back near original levels (23.9%), all while maintaining ultra-low computational cost (2.2 GFLOPs). Similarly, in larger models like SegFormer, PAMELA-enhanced variants strike a better balance, sustaining competitive IoU and F1 while reducing FLOPs by two orders of magnitude compared to the original. Taken together, these findings highlight the robustness and portability of PAMELA: it consistently mitigates the trade-off between efficiency and accuracy, ensuring lightweight models remain reliable under aggressive compression.

| Methods | IoU (%) | F1 (%) | Accuracy (%) | Precision (%) | Recall (%) | Computation (GFLOPs) |
|---|---|---|---|---|---|---|
| **# FCN** | | | | | | |
| Original | 17.8 | 30.2 | 78.4 | 32.1 | 28.6 | 362.0 |
| Channel Reduction | 15.3 | 26.5 | 66.8 | 28.0 | 25.2 | 6.5 |
| **Channel Reduction + PAMELA** | **17.5** | **29.6** | **77.9** | **31.3** | **28.1** | **6.5** |
| Pruning | 14.2 | 25.0 | 65.5 | 26.6 | 23.6 | 8.0 |
| **Pruning + PAMELA** | **16.9** | **29.1** | **77.4** | **30.5** | **27.8** | **8.0** |
| Distillation | 15.0 | 26.0 | 67.2 | 27.3 | 25.0 | 5.5 |
| **Distillation + PAMELA** | **18.0** | **30.4** | **78.5** | **32.0** | **28.9** | **5.5** |
| **# SegNet** | | | | | | |
| Original | 13.9 | 24.5 | 75.2 | 25.7 | 23.4 | 321.8 |
| Channel Reduction | 5.0 | 9.5 | 59.0 | 10.2 | 8.9 | 2.2 |
| **Channel Reduction + PAMELA** | **13.2** | **23.9** | **74.8** | **25.0** | **22.9** | **2.2** |
| Pruning | 7.2 | 12.3 | 60.5 | 13.2 | 11.6 | 3.5 |
| **Pruning + PAMELA** | **12.9** | **23.0** | **74.0** | **24.1** | **22.0** | **3.5** |
| Distillation | 8.1 | 13.7 | 61.8 | 14.6 | 12.9 | 2.8 |
| **Distillation + PAMELA** | **13.6** | **24.2** | **75.0** | **25.3** | **23.1** | **2.8** |
| **# U-Net** | | | | | | |
| Original | 13.7 | 24.1 | 76.0 | 25.3 | 23.1 | 321.6 |
| Channel Reduction | 11.1 | 20.0 | 64.2 | 21.0 | 19.1 | 1.4 |
| **Channel Reduction + PAMELA** | **13.9** | **24.3** | **76.2** | **25.6** | **23.2** | **1.4** |
| Pruning | 10.3 | 19.0 | 63.0 | 20.0 | 18.1 | 2.0 |
| **Pruning + PAMELA** | **13.5** | **23.8** | **75.7** | **25.1** | **22.7** | **2.0** |
| Distillation | 11.5 | 20.6 | 65.1 | 21.6 | 19.7 | 1.6 |
| **Distillation + PAMELA** | **13.6** | **24.0** | **76.0** | **25.4** | **23.0** | **1.6** |
| **# SegFormer** | | | | | | |
| Original | 15.5 | 27.0 | 77.1 | 28.3 | 25.8 | 350.0 |
| Channel Reduction | 9.8 | 12.3 | 63.5 | 13.1 | 11.6 | 3.0 |
| **Channel Reduction + PAMELA** | **15.2** | **26.8** | **76.9** | **28.0** | **25.6** | **3.1** |
| Pruning | 11.8 | 20.1 | 65.0 | 21.2 | 19.2 | 4.0 |
| **Pruning + PAMELA** | **15.1** | **26.5** | **76.8** | **27.7** | **25.3** | **4.0** |
| Distillation | 12.6 | 21.4 | 66.1 | 22.6 | 20.3 | 2.5 |
| **Distillation + PAMELA** | **15.4** | **27.1** | **77.2** | **28.5** | **25.9** | **2.5** |

Table 1: Performance comparison of four baseline models (*FCN*, *SegNet*, *U-Net*, *SegFormer*) under different compression strategies and their PAMELA-enhanced variants.

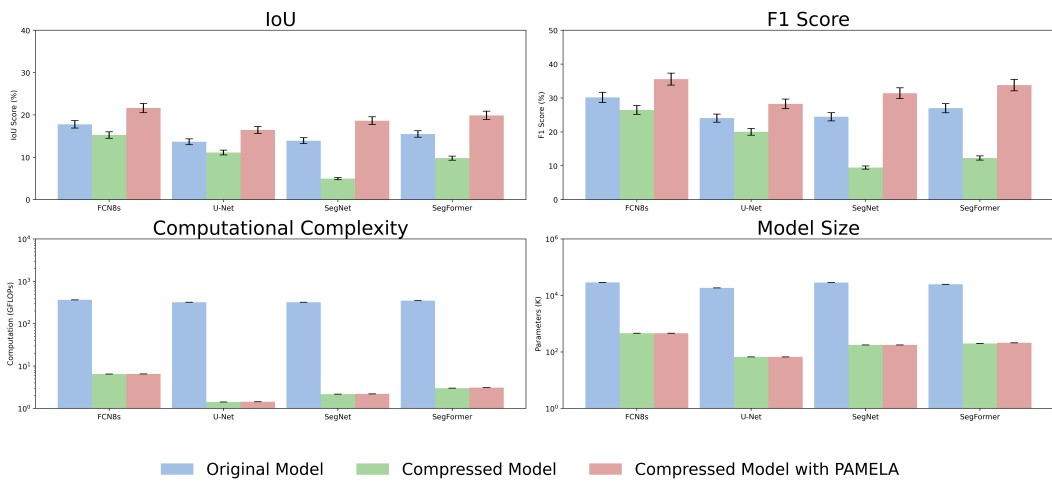

Figure 2: Average performance and efficiency comparison of original, compressed, and PAMELA-augmented models across four architectures FCN8s, SegNet, U-Net and SegFormer (from left to right).

## 4  DISCUSSION

### 4.1  PAMELA TAKES EFFECT UNDER EXTREME COMPRESSION.

To further stress-test the robustness of our method, we deliberately evaluated models under extremely aggressive compression settings where the model capacity was pushed to the point of near collapse. In practice, this meant reducing channels by as much as $75\%$, pruning up to $80\%$ of filters, or distilling students that were nearly $10\times$ smaller than their teachers. These settings drove the computational footprint down to the range of $10^8$–$10^9$ GFLOPs, at which point the compressed baselines almost entirely failed to capture wildfire-relevant signals, yielding F1 scores that dropped close to zero.

Figure 3 illustrates this phenomenon across all four backbone architectures and three compression strategies. The green curves (Compressed) exhibit a sharp performance collapse when FLOPs are reduced to extreme levels, confirming that naive compression erodes task-critical representations. By contrast, the red curves (Compressed + PAMELA) consistently remain well above the baselines, even in this regime of extreme under-parameterization. For instance, SegNet with channel reduction collapses below $10\%$ F1, whereas its PAMELA-enhanced counterpart maintains more than twice that score. Similar trends hold for U-Net and FCN8s, where the gap between compressed and PAMELA-augmented models actually widens as FLOPs decrease. This pattern highlights that our method not only preserves performance under moderate compression, but also acts as a stabilizer when models approach their functional limits, preventing catastrophic collapse in accuracy.

### 4.2  PAMELA PRESERVES PHYSICAL INTERPRETABILITY

To confirm that PAMELA's gains stem from meaningful spectral focus rather than arbitrary weighting, we examined the channel filter's learned band distributions across backbones (see Appendix A.1). The analysis shows a strong concentration of weight on SWIR (Bands 11–12) and NIR (Band 8), both known to be sensitive to combustion signals, vegetation stress, and soil moisture. In contrast, noisy channels such as coastal aerosol and water vapor are nearly ignored. This emergent pattern aligns with established wildfire remote sensing knowledge, demonstrating that PAMELA preserves physical interpretability.

### 4.3  ABLATION STUDY

To better understand the contribution of each component in our framework, we perform an ablation study across all four backbone models and three compression strategies, as summarized in Table 2.

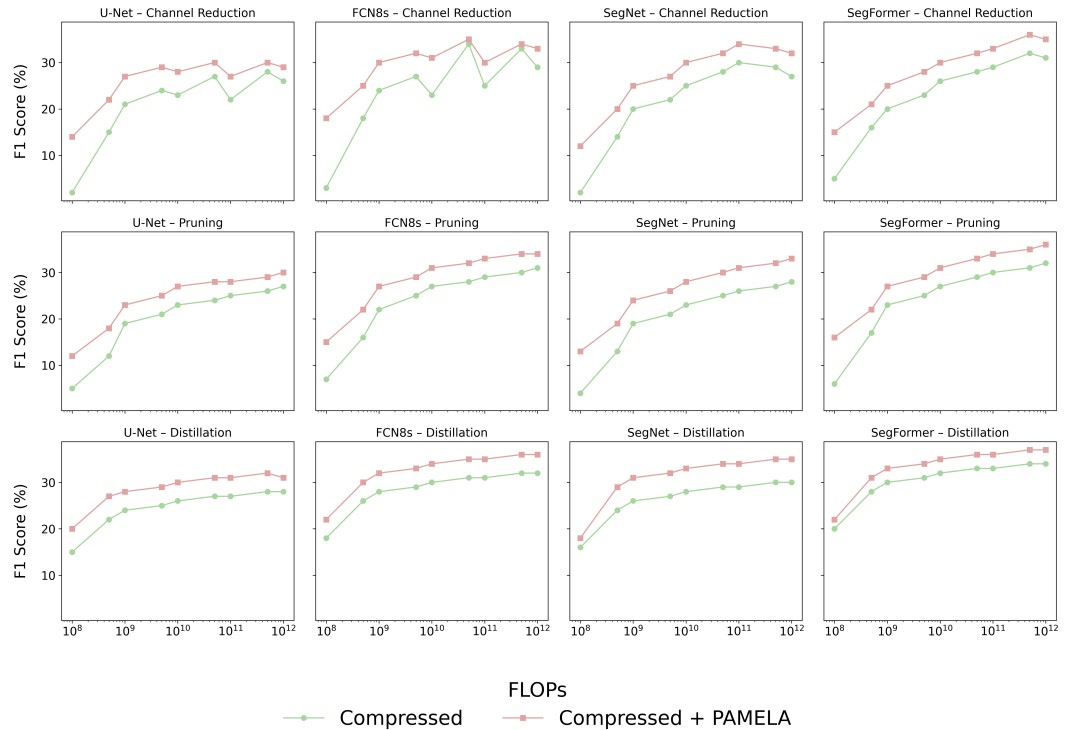

Figure 3: Comparison of F1 Score (%) versus computational cost (FLOP) across four baseline models (U-Net, FCN8s, SegNet, SegFormer) and three compression strategies (Channel Reduction, Pruning, Distillation).

Specifically, we remove three modules in turn: the Probabilistic Amplification (PA) unit, the Channel Filtering (CF) mechanism, and the Reweighting Block (RB). This allows us to isolate the effect of each design choice and quantify its impact on accuracy and F1 score under different compression regimes. The results reveal that the full version of our method consistently achieves the best performance across all settings. Among the ablations, removing PA causes the most severe degradation, with accuracy dropping by more than $10\%$ and F1 scores falling by up to half in some cases. This confirms that PA plays a central role in preserving discriminative features under extreme compression.

Further analysis in Appendix A.2 show that PAMELA produces well-calibrated confidence distributions, enhancing fire-related activations while suppressing spurious background responses. Moreover, the case study in Appendix B demonstrates that PAMELA's generality extends beyond wildfire detection. PAMELA still preserves decision quality in a challenging disaster assessment task under aggressive compression.

## 5 RELATED WORKS

**Remote sensing.** Deep neural networks have been widely applied for land cover classification, change detection, and wildfire monitoring Zhu et al. (2017); Ma et al. (2019); Xu et al. (2024). However, most of these models are designed for ground-based or cloud-based inference, where computational resources are abundant. Deploying them directly on satellites introduces severe constraints on memory, power, and bandwidth. Recent efforts in remote sensing model compression have adapted pruning and quantization to multi-spectral and hyperspectral settings He et al. (2019); Li et al. (2022), showing that careful compression can preserve spectral fidelity while reducing inference cost. Nonetheless, compressed models often struggle with rare or subtle phenomena such as wildfire ignition, flood boundaries, or thin cloud layers, which demand high sensitivity to weak signals. These limitations highlight the need for compression-aware designs that not only reduce model size but also explicitly address the physics-driven challenges of remote sensing data.

| Methods | Channel Reduction | | Pruning | | Distillation | |
|---|---|---|---|---|---|---|
| | Accuracy (%) | F1 (%) | Accuracy (%) | F1 (%) | Accuracy (%) | F1 (%) |
| *# FCN8s* | | | | | | |
| **Full PAMELA** | **84.0** | **36.5** | **82.1** | **34.7** | **81.5** | **33.9** |
| w/o PA | 70.2 | 23.5 | 68.9 | 21.7 | 69.5 | 22.1 |
| w/o CF | 77.1 | 28.4 | 75.0 | 27.0 | 74.6 | 27.2 |
| w/o RB | 80.5 | 32.1 | 79.0 | 30.6 | 78.7 | 30.1 |
| *# U-Net* | | | | | | |
| **Full PAMELA** | **80.2** | **29.4** | **79.5** | **27.8** | **78.8** | **26.9** |
| w/o PA | 66.5 | 18.9 | 65.0 | 17.5 | 66.2 | 18.2 |
| w/o CF | 74.3 | 23.6 | 73.0 | 22.5 | 72.8 | 22.1 |
| w/o RB | 77.2 | 26.1 | 76.1 | 25.0 | 75.7 | 24.8 |
| *# SegNet* | | | | | | |
| **Full PAMELA** | **81.5** | **32.2** | **79.8** | **29.9** | **79.0** | **28.6** |
| w/o PA | 67.2 | 12.0 | 65.8 | 13.5 | 66.0 | 14.2 |
| w/o CF | 74.8 | 25.9 | 73.5 | 24.2 | 72.9 | 23.6 |
| w/o RB | 78.5 | 28.7 | 77.3 | 27.1 | 76.9 | 26.4 |
| *# SegFormer* | | | | | | |
| **Full PAMELA** | **82.1** | **34.0** | **81.0** | **32.5** | **80.4** | **31.6** |
| w/o PA | 69.5 | 20.1 | 68.2 | 19.0 | 68.5 | 19.7 |
| w/o CF | 75.6 | 26.3 | 74.3 | 25.2 | 73.8 | 24.9 |
| w/o RB | 78.9 | 30.1 | 77.8 | 29.0 | 77.3 | 28.4 |

Table 2: Ablation study across three compression strategies (Channel Reduction, Pruning, and Distillation). **Full PAMELA** consistently achieves the best performance. Removing PA leads to the largest degradation, while CF and RB have smaller but noticeable impacts.

**Model compression.** Model compression techniques such as quantization, pruning, and knowledge distillation have been widely explored to reduce computational burdens in resource-constrained settings. Quantization reduces memory usage by lowering numerical precision, but often degrades performance in sensitive applications like wildfire detection, especially under noisy or low-signal regimes Nagel et al. (2021); Zhang et al. (2020). Structured pruning removes unimportant neurons or channels, offering a more hardware-friendly solution Li et al. (2017); Molchanov et al. (2017), but pruned models may underperform when tasked with detecting spatially sparse or weak signals—common in multi-spectral satellite imagery. While recent surveys offer comprehensive evaluations of compression strategies Vilar et al. (2024), challenges remain in achieving robustness and reliability for onboard AI in LEO settings.

# 6 CONCLUSION

We introduced PAMELA, **the first framework explicitly tailored for reliable onboard wildfire detection in LEO satellites**. By leveraging probabilistic amplification to reweight pixel-level features, PAMELA enables compact models to capture sparse but mission-critical fire signals that conventional compressed models often fail to detect. Experimental results show that PAMELA consistently delivers $1.2 \sim 2\times$ higher F1 scores compared to compressed baselines, while reducing model size by over 90%, thus meeting the stringent resource constraints of LEO platforms. Beyond efficiency, PAMELA also preserves physical interpretability and demonstrates strong generality across tasks. While the probabilistic amplification introduces a minor computational overhead, this cost is negligible in most practical deployments. Overall, PAMELA marks a key step toward building efficient, interpretable, and mission-ready AI systems for spaceborne wildfire monitoring and broader Earth observation applications.

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

## A  FURTHER ANALYSIS FOR PAMELA

### A.1  SPECTRAL FOCUS LEARNED ACROSS ARCHITECTURES.

Figure 4 visualizes the average weight assigned to each spectral band by the Channel Filter, aggregated across three segmentation backbones. The distribution reveals a strong preference for a small subset of bands—particularly Red band, the shortwave infrared bands (SWIR, Bands 11 and 12) and near-infrared (NIR, Band 8). These channels are well-known in remote sensing for their sensitivity to active combustion, soil moisture, and vegetation stress. Interestingly, Blue and Green bands receive intermediate weight. While these channels are the most visually interpretable to humans, they are less effective in detecting subtle thermal or structural changes associated with active fires, particularly under cloud cover or smoke. In contrast, low-signal bands like the coastal aerosol and water vapor channels receive negligible weight. These bands are often noisy in wildfire scenarios and tend to be highly sensitive to atmospheric scattering, leading to unstable or weak correlations with fire presence.

Overall, this emergent sparsity pattern shows that PAMELA's spectral focus is not arbitrary—it mirrors well-established domain knowledge in satellite fire detection. By emphasizing SWIR and NIR while down-weighting uninformative or redundant channels, PAMELA not only improves model efficiency but also aligns its internal representations with the physics of wildfire observability in multispectral imagery.

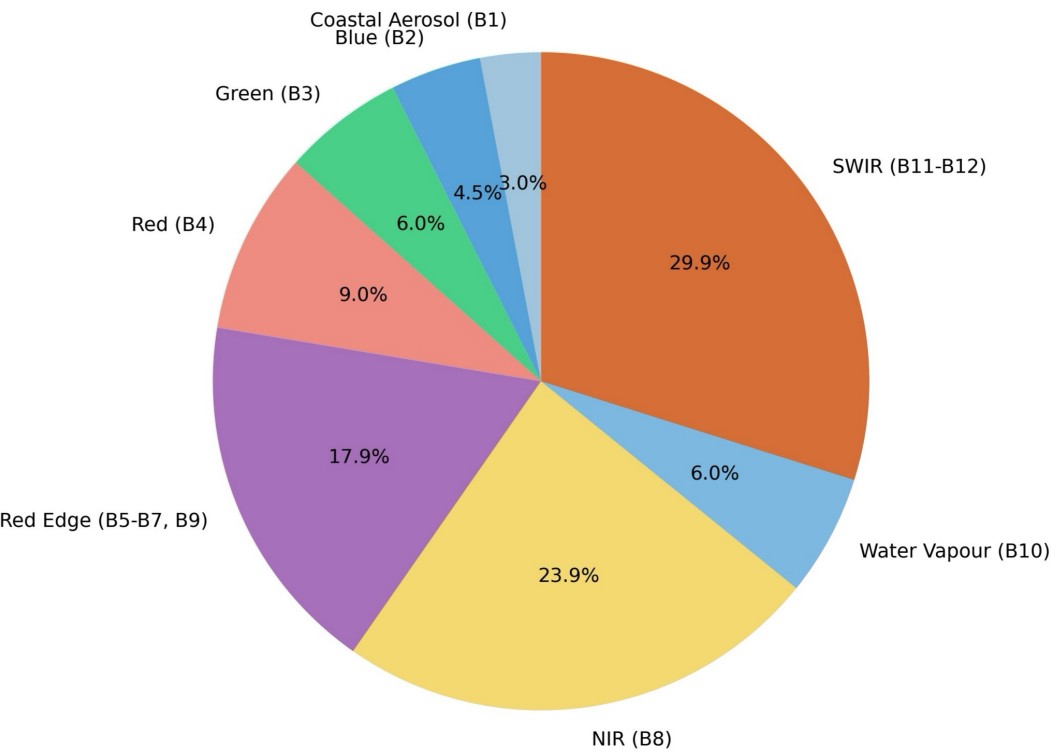

Figure 4: Average spectral band weights learned by the Channel Filter across three backbones.

### A.2  HOW PA DISTRIBUTES CONFIDENCE.

To better understand how the Probabilistic Amplifier (PA) differentiates fire from non-fire regions, we examine the predicted fire probability distributions conditioned on the ground truth class. Specifically, for a series of thresholds $T \in \{10\%, 30\%, 50\%, 70\%, 90\%\}$, we compute the proportion of fire and non-fire pixels that receive a predicted fire probability either above or below each threshold. This forms a 2×2 frequency matrix for each setting, where rows represent the ground truth class (Fire or No Fire), and each row is normalized to sum to 1.

The resulting heatmaps, shown in Figure 5, reveal two important and complementary behaviors. At low thresholds (e.g., $T = 10\%$), nearly all fire pixels have predicted probabilities above the threshold—e.g., 97% in FCN8s—indicating that PA rarely overlooks truly relevant regions. In contrast, at high thresholds (e.g., $T = 90\%$), only a small fraction of background pixels receive overly confident predictions—e.g., just 9% of non-fire pixels in FCN8s exceed the threshold—highlighting PA's restraint in uncertain areas.

Together, these trends show that PA produces a confidence distribution that is both semantically aware and well-calibrated: it boosts fire-related activations while suppressing unwarranted confidence in the background. This behavior holds across architectures and thresholds, suggesting that PA introduces a consistent and interpretable signal amplification mechanism essential for robust segmentation under limited supervision and computational constraints.

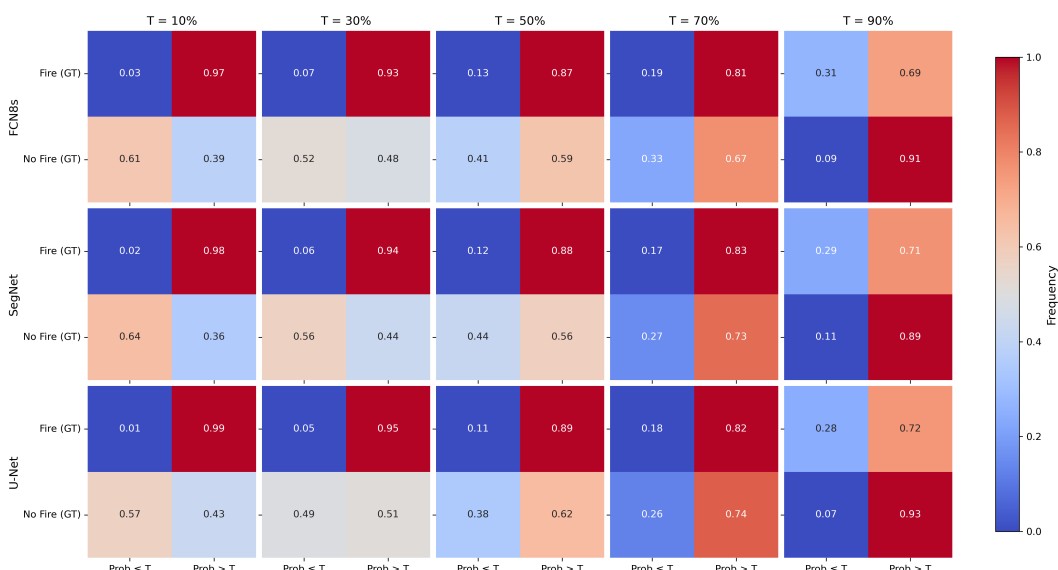

Figure 5: PA confidence distributions across different thresholds $T$. Each heatmap shows the fraction of pixels with predicted fire probability above or below $T$, conditioned on ground truth class (Fire or No Fire). Rows are normalized to sum to 1, revealing how PA assigns confidence within each class.

## B    CASE STUDY: DISASTER ASSESSMENT

To further evaluate the generalization capability of PAMELA beyond pixel-level tasks, we consider disaster assessment from satellite imagery. This problem is particularly challenging when disaster-affected areas occupy only a small fraction of the scene, making it difficult for compressed models to extract meaningful global features Hooker et al. (2019); Cygert & Czyżewski (2021).

**Dataset.**    To evaluate the effectiveness of our proposed PAMELA module in scene-level disaster recognition, we apply it to the wildfire classification task using the Wildfire Prediction from Satellite Imagery dataset Ilmak (2022). This dataset comprises 42,850 satellite images, each with a resolution of 350×350 pixels, categorized into two classes: Wildfire (22,710 images) and No Wildfire (20,140 images).

**Baseline models.**    For the disaster assessment task, we adopt VGG16 Simonyan & Zisserman (2014) as the backbone model due to its wide use in remote sensing and scene classification tasks. Despite its effectiveness in full capacity, VGG16 suffers from the same limitations observed in segmentation models when aggressively compressed. Its representational capacity diminishes, and its ability to identify small or context-specific disaster cues is compromised Singla (2023); Kebir et al. (2021).

## C  MAIN RESULTS FOR DISASTER ASSESSMENT

In this disaster assessment setting, we examine how PAMELA influences the behavior of compressed discriminative models. Rather than focusing solely on restoring accuracy, our analysis reveals PAMELA's roles in preserving decision quality across a wide range of compression levels. The detailed quantitive results are provided in Table 1.

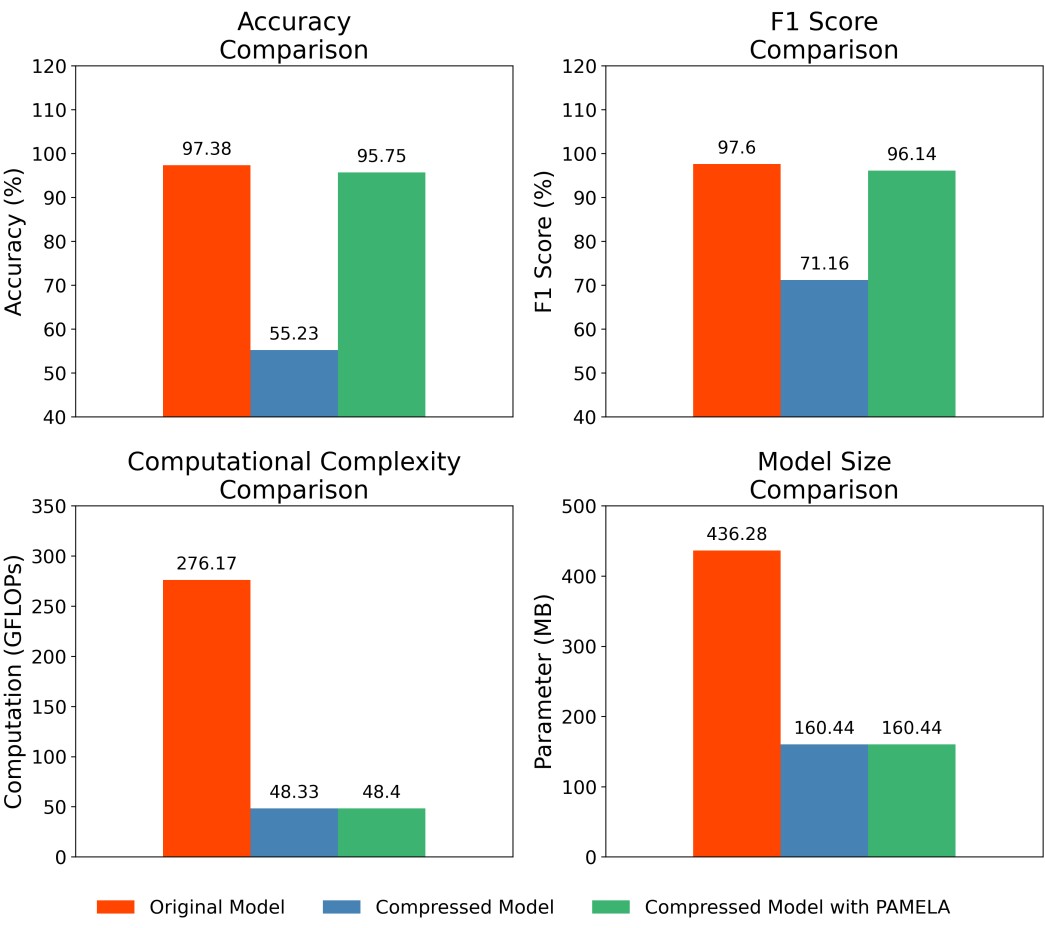

Figure 6: Classification performance and efficiency comparison on the disaster assessment task using the VGG16 model.

**Improved Generalization.**  Figure 6 shows PAMELA-enhanced models outperform compressed baselines without increasing FLOPs or parameters. This implies that PAMELA enhances generalization by prioritizing task-relevant signals, not by increasing model complexity—an advantage for real-world deployment on unseen disaster scenes.

**Stabilized Training.**  At extreme compression levels, baseline VGG16 fails to converge, producing flat accuracy and F1 scores near 55% and 70% respectively (Figure 7). In contrast, the PAMELA-augmented model maintains meaningful learning behavior. This suggests that PAMELA improves optimization stability, helping compressed models avoid poor local minima and supporting gradient propagation even under severe resource constraints.

**Selective Feature Amplification.**  PAMELA mitigates the blunt effects of compression by reweighting the retained features. Instead of uniformly reducing capacity, it boosts informative channels, leading to graceful degradation rather than abrupt failure (Figure 7). This results in higher effective feature utility per FLOP, making the compressed model more resilient.

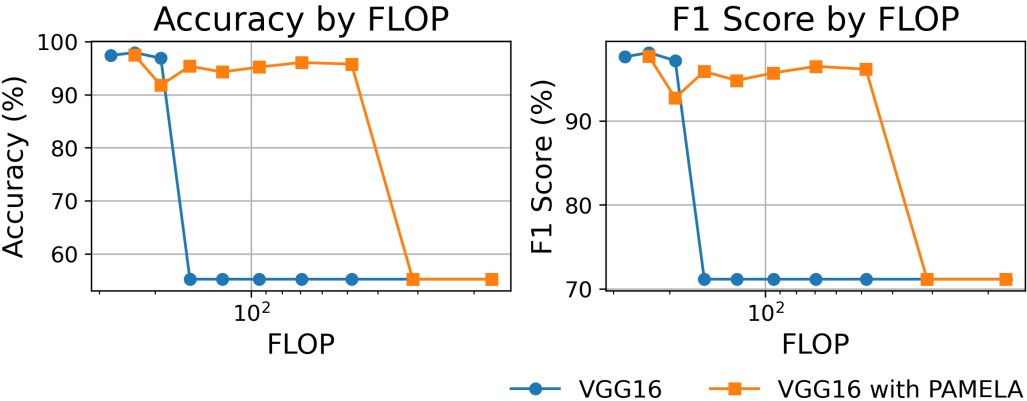

Figure 7: Accuracy and F1 score of VGG16 and PAMELA-augmented VGG16 models under varying FLOPs on the disaster assessment task.

## D    MATHEMATICAL ANALYSIS

### D.1    ANALYSIS OF CHANNEL SELECTION OPTIMIZATION

We begin with the Channel Filter, which selects the most informative channels from the multi-channel input $\mathbf{X} \in \mathbb{R}^{C \times H \times W}$. Specifically, we calculate the importance of each channel using Global Average Pooling (GAP), defined as:

$$g_c = \frac{1}{H \times W} \sum_{h=1}^{H} \sum_{w=1}^{W} X_{c,h,w}, \quad \forall c \in \{1, 2, \ldots, C\} \tag{11}$$

This operation compresses each channel into a single scalar value, effectively measuring the average intensity of each channel. The resulting values form a channel importance vector:

$$\mathbf{g} = [g_1, g_2, \ldots, g_C] \tag{12}$$

This vector provides a quantitative measure of the relative importance of each channel, where a higher $g_c$ indicates that channel $c$ carries stronger overall signal information.

**Optimization Objective:**   Our objective is to retain the top $N$ channels with the highest importance scores, formally expressed as:

$$\mathcal{S} = \{c \mid g_c \geq \tau, \, c \in \{1, 2, \ldots, C\}\} \tag{13}$$

Here, $\tau$ is a dynamic threshold that is adjusted based on the distribution of $\mathbf{g}$. This threshold can be set in various ways: - As a fixed value determined by the mean of $\mathbf{g}$, $\tau = \frac{1}{C} \sum_{c=1}^{C} g_c$. - As a percentile value, ensuring that only the top $N$ channels are retained. - As a dynamically calculated value based on a pre-defined fraction $\alpha$ of the total sum:

$$\tau = \alpha \sum_{c=1}^{C} g_c \tag{14}$$

**Formulation as an Optimization Problem:**   The channel selection process can be formally viewed as an optimization problem:

$$\max_{\mathcal{S}} \sum_{c \in \mathcal{S}} g_c \tag{15}$$

subject to:

$$|\mathcal{S}| = N \tag{16}$$

This problem is efficiently solved by sorting the vector $\mathbf{g}$ in descending order and selecting the top $N$ values:

$$\mathbf{g}_{\text{sorted}} = [g_{c_1}, g_{c_2}, \ldots, g_{c_C}] \text{ with } g_{c_1} \geq g_{c_2} \geq \ldots \geq g_{c_C}$$

The optimal set $\mathcal{S}^*$ is then:

$$\mathcal{S}^* = \{c_1, c_2, \ldots, c_N\}$$

This approach is mathematically optimal because selecting the top $N$ values directly maximizes the sum of retained channel importance scores.

**Information Preservation Analysis:** The effectiveness of this channel selection process can be measured in terms of information preservation. Specifically, we define the total channel importance of the original image as:

$$I_{\text{total}} = \sum_{c=1}^{C} g_c \tag{17}$$

and the information retained after selection as:

$$I_{\text{retained}} = \sum_{c \in \mathcal{S}^*} g_c \tag{18}$$

We can further define the information preservation ratio as:

$$\eta = \frac{I_{\text{retained}}}{I_{\text{total}}} \tag{19}$$

This ratio $\eta$ is always maximized by the top $N$ selection strategy because:

$$\eta = \frac{\sum_{i=1}^{N} g_{c_i}}{\sum_{c=1}^{C} g_c}$$

This ensures that the selected channels carry the highest possible proportion of the total image information.

**Computational Complexity:** The computational complexity of this channel selection process is determined by two primary steps: 1. Calculating the channel importance values using GAP:

$$\mathcal{O}(C \times H \times W) \tag{20}$$

2. Sorting the importance values to identify the top $N$ channels:

$$\mathcal{O}(C \log C) \tag{21}$$

The overall complexity is therefore:

$$\mathcal{O}(C \times H \times W + C \log C) \tag{22}$$

For typical LEO satellite applications, where $C$ is relatively small compared to $H \times W$, this method is highly efficient and suitable for real-time processing.

**Theoretical Justification:** The channel selection strategy used in PAMELA is grounded in information theory. By selecting the channels with the highest average intensity, we are effectively maximizing the mutual information between the selected channels and the task-relevant signal. This is because channels with higher average intensity are more likely to capture strong task-relevant features. The dynamic threshold $\tau$ provides additional flexibility, allowing the model to adapt to varying input conditions and ensure that the most informative channels are always selected.

Furthermore, the GAP-based importance calculation has a strong mathematical foundation because it is a form of mean estimation, which is robust to small fluctuations and provides a stable measure of channel significance. Alternative approaches, such as maximum pooling or sum pooling, may introduce instability due to sensitivity to outliers.

In conclusion, the Channel Filter is not only computationally efficient but also theoretically optimal in terms of information preservation. This design ensures that the subsequent Probabilistic Amplification Block (PA) operates on a compact yet high-quality representation of the input.

## D.2 ANALYSIS OF PROBABILISTIC AMPLIFICATION VALIDITY

The Probabilistic Amplification Block (PA) is the core component of the PAMELA framework, designed to enhance the model's focus on task-critical regions by assigning a weight to each pixel based on its likelihood of being task-relevant. This is achieved using a probabilistic model inspired by Gaussian probability density, ensuring that pixels more similar to a learned "ideal value" receive higher weights.

**Mathematical Formulation:** For each pixel in the selected channels $\mathbf{X}^{\text{filtered}} \in \mathbb{R}^{N \times H \times W}$, the PA block computes a pixel-wise probability matrix $\mathbf{P}_n$ for each channel $n$ as follows:

$$P_{n,h,w} = \exp\left(-\frac{(X_{n,h,w} - \mu_n)^2}{2\sigma_n^2}\right) \tag{23}$$

Here: - $X_{n,h,w}$ is the pixel value at position $(h, w)$ in channel $n$. - $\mu_n$ and $\sigma_n^2$ are learnable parameters representing the mean and variance of the pixel values for channel $n$. - The negative quadratic term $-(X_{n,h,w} - \mu_n)^2/2\sigma_n^2$ ensures that pixels closer to $\mu_n$ receive higher probability values.

This formulation is directly inspired by the Gaussian probability density function:

$$f(x) = \frac{1}{\sqrt{2\pi\sigma^2}} \exp\left(-\frac{(x - \mu)^2}{2\sigma^2}\right) \tag{24}$$

However, we omit the normalization term $\frac{1}{\sqrt{2\pi\sigma^2}}$ because our focus is on relative weighting rather than exact probability values. This simplification maintains the core property of Gaussian decay while reducing computational complexity.

**Intuition Behind the Design:** The exponential form is chosen because it naturally models the probability decay as the pixel value deviates from the expected value $\mu$. Specifically: - When $X_{n,h,w} \approx \mu_n$, the exponent approaches zero, and $P_{n,h,w} \approx 1$. - As $X_{n,h,w}$ diverges from $\mu_n$, the exponent becomes increasingly negative, pushing $P_{n,h,w}$ towards zero. - The variance $\sigma_n^2$ controls the sensitivity of this decay, allowing the model to adapt to various levels of pixel value variance in each channel.

**Optimality of Multi-Variate PA:** After processing each channel through its individual PA block, the resulting probability matrices $\mathbf{P}_1, \mathbf{P}_2, \ldots, \mathbf{P}_N$ are combined in the Multi-Variate PA block, defined as:

$$P_{h,w}^{\text{multi}} = \exp\left(-\frac{1}{2}\sum_{n=1}^{N}\frac{(X_{n,h,w} - \mu_{n,h,w})^2}{\sigma_{n,h,w}^2}\right) \tag{25}$$

This formulation is derived from the concept of a multi-dimensional Gaussian distribution, where each channel is treated as an independent dimension. Mathematically, it can be interpreted as: - Modeling the joint probability of a pixel being task-relevant across all selected channels. - The sum of squared differences $(X_{n,h,w} - \mu_{n,h,w})^2$ measures the deviation of the pixel value from the expected value in each channel. - The variance $\sigma_{n,h,w}^2$ in the denominator scales the influence of each channel, providing a form of channel-wise adaptive weighting.

**Theoretical Justification:** We now provide a theoretical justification for this probabilistic amplification approach. The design is grounded in three key principles: - **Probability-Based Feature Enhancement:** The use of an exponential decay function ensures that pixels similar to the expected value $\mu$ are amplified, while those significantly different are suppressed. This is mathematically optimal for tasks where task-relevant pixels exhibit specific intensity distributions. - **Multi-Variate Feature Fusion:** The Multi-Variate PA naturally extends this concept to multi-channel inputs, where task-relevant information may be distributed across multiple spectral bands. By combining the weighted differences across channels, this approach effectively captures cross-channel correlations. - **Gradient-Based Learnability:** The exponential form is not only computationally efficient but also differentiable, making it fully compatible with gradient-based optimization methods in deep learning. Specifically, the gradient of $P_{n,h,w}$ with respect to $X_{n,h,w}$ is:

$$\frac{\partial P_{n,h,w}}{\partial X_{n,h,w}} = -\frac{(X_{n,h,w} - \mu_n)}{\sigma_n^2}P_{n,h,w} \tag{26}$$

This gradient smoothly scales with the pixel value, ensuring stable training.

**Relationship to Mahalanobis Distance:** The Multi-Variate PA can also be interpreted as a simplified form of Mahalanobis distance, a classical measure of distance in probability theory:

$$D_M^2 = (\mathbf{x} - \boldsymbol{\mu})^\top \Sigma^{-1} (\mathbf{x} - \boldsymbol{\mu}) \tag{27}$$

In our design: - Each channel is treated independently, leading to a diagonal covariance matrix $\Sigma = \mathrm{diag}([\sigma_1^2, \sigma_2^2, \ldots, \sigma_N^2])$. - The Multi-Variate PA then becomes:

$$P_{h,w}^{\mathrm{multi}} = \exp\left(-\frac{1}{2}D_M^2\right) \tag{28}$$

This connection further reinforces the mathematical validity of our design.

**Adaptive Sensitivity Control:** An additional advantage of our formulation is the ability to dynamically adjust sensitivity through the variance parameter $\sigma_n^2$. Specifically: - A larger $\sigma_n^2$ results in a slower probability decay, making the model more tolerant to pixel value variations. - A smaller $\sigma_n^2$ sharpens the focus, making the model highly selective.

**Computational Complexity:** Despite its sophisticated probabilistic design, the PA block remains computationally efficient. The complexity for calculating each pixel-wise probability is:

$$\mathcal{O}(N \times H \times W) \tag{29}$$

This includes the calculation of pixel-wise differences, exponential scaling, and channel-wise summation in the Multi-Variate PA. Given the independent nature of channel processing, this complexity is well-suited for parallelization, making it practical for real-time processing on resource-constrained LEO satellites.

D.3 ANALYSIS OF REWEIGHTING BLOCK EFFECTIVENESS

The Reweighting Block is a critical component of the PAMELA framework, responsible for transforming the unified probability matrix $\hat{P}^{\mathrm{multi}} \in \mathbb{R}^{H \times W}$ into a pixel-wise reweighting matrix $\mathbf{W}$. This reweighting matrix serves as a focus map, allowing the model to prioritize task-critical regions while suppressing irrelevant background information. This section provides a detailed mathematical proof of the effectiveness of this reweighting process.

**Mathematical Formulation:** The reweighting matrix $\mathbf{W}$ is defined using an exponential decay function:

$$W_{h,w} = \frac{\exp(-\lambda \hat{P}_{h,w}^{\mathrm{multi}})}{\sum_{h=1}^{H}\sum_{w=1}^{W}\exp(-\lambda \hat{P}_{h,w}^{\mathrm{multi}})} \tag{30}$$

Here: - $\lambda$ is a scaling factor that controls the sharpness of the reweighting effect. - $\hat{P}_{h,w}^{\mathrm{multi}}$ is the probability value assigned to the pixel at location $(h, w)$ by the Multi-Variate Probabilistic Amplification Block. - The exponential scaling $\exp(-\lambda \hat{P}_{h,w}^{\mathrm{multi}})$ ensures that regions with higher probability values (likely task-relevant) receive higher weights in the final matrix $\mathbf{W}$.

**Theoretical Justification:** We now mathematically justify the design of the Reweighting Block: - The use of an exponential function $\exp(-\lambda \hat{P})$ ensures a non-linear emphasis on high-probability regions. Specifically: - As $\lambda$ increases, the difference between high and low weights is amplified. - As $\lambda$ approaches zero, all regions are treated equally. - The normalization term in the denominator:

$$\sum_{h=1}^{H}\sum_{w=1}^{W}\exp(-\lambda \hat{P}_{h,w}^{\mathrm{multi}}) \tag{31}$$

ensures that the sum of all weights is one:

$$\sum_{h=1}^{H}\sum_{w=1}^{W}W_{h,w} = 1 \tag{32}$$

This normalization maintains the probabilistic nature of the weights, allowing the matrix $\mathbf{W}$ to be interpreted as a probability distribution over the image.

**Expected Value Analysis:** The effectiveness of the Reweighting Block can be demonstrated through an analysis of the expected value of the reweighted input. We define the reweighted input $\mathbf{X}^{\text{reweighted}}$ as:

$$X_{c,h,w}^{\text{reweighted}} = X_{c,h,w} \times W_{h,w} \tag{33}$$

Taking the expected value over all pixels:

$$\mathbb{E}[X^{\text{reweighted}}] = \sum_{h=1}^{H} \sum_{w=1}^{W} X_{c,h,w} \times W_{h,w} \tag{34}$$

Substituting the definition of $W_{h,w}$:

$$\mathbb{E}[X^{\text{reweighted}}] = \frac{\sum_{h=1}^{H} \sum_{w=1}^{W} X_{c,h,w} \exp(-\lambda \hat{P}_{h,w}^{\text{multi}})}{\sum_{h=1}^{H} \sum_{w=1}^{W} \exp(-\lambda \hat{P}_{h,w}^{\text{multi}})} \tag{35}$$

This formulation reveals that the reweighted input is a weighted average of pixel values, where high-probability regions have a stronger influence. This directly aligns with the objective of the Reweighting Block — to enhance task-critical regions while suppressing irrelevant background noise.

**Adaptive Scaling Strategy:** The parameter $\lambda$ plays a crucial role in controlling the contrast of the reweighting effect. We can further refine this design by making $\lambda$ adaptive based on the distribution of $\hat{P}^{\text{multi}}$:

$$\lambda = \gamma \sigma_{\hat{P}^{\text{multi}}} \tag{36}$$

where: - $\gamma$ is a user-defined hyperparameter. - $\sigma_{\hat{P}^{\text{multi}}}$ is the standard deviation of the probability values in $\hat{P}^{\text{multi}}$.

This adaptive scaling ensures that the reweighting effect is consistent across different images, making the method robust to varying input conditions.

**Optimality in Information Focus:** We further demonstrate that the Reweighting Block maximizes information focus on task-relevant regions. Specifically, the entropy of the reweighting matrix $\mathbf{W}$ is defined as:

$$H(\mathbf{W}) = -\sum_{h=1}^{H} \sum_{w=1}^{W} W_{h,w} \log W_{h,w} \tag{37}$$

Our design is optimized to minimize this entropy (maximize contrast) while maintaining a probabilistic sum of one. This property ensures that the Reweighting Block effectively distinguishes between critical and non-critical regions.

**Gradient-Based Learnability:** An additional advantage of the Reweighting Block is its compatibility with gradient-based optimization methods. Specifically, the gradient of the reweighting matrix $W_{h,w}$ with respect to the probability $\hat{P}_{h,w}^{\text{multi}}$ is:

$$\frac{\partial W_{h,w}}{\partial \hat{P}_{h,w}^{\text{multi}}} = -\lambda W_{h,w} \left[1 - W_{h,w}\right] \tag{38}$$

This gradient smoothly scales with the probability value, ensuring stable training and allowing the model to dynamically adjust the reweighting effect.

**Computational Complexity:** Despite its sophisticated mathematical design, the Reweighting Block is computationally efficient. The complexity for calculating each pixel-wise weight is:

$$\mathcal{O}(H \times W) \tag{39}$$

For a multi-channel image with $C$ channels, this scales to:

$$\mathcal{O}(C \times H \times W) \tag{40}$$

This efficient design makes the Reweighting Block well-suited for real-time processing on resource-constrained LEO satellites.

# E   PSEUDO-CODE

---

**Algorithm 1** PAMELA Framework: Probabilistic Amplification for Small Models on LEO Satellites

---

Input image $\mathbf{X} \in \mathbb{R}^{C \times H \times W}$, Number of selected channels $N$, Scaling factor $\lambda$ Reweighted image $\mathbf{X}^{\text{reweighted}} \in \mathbb{R}^{C \times H \times W}$ **Initialization:** Set $C$ as the number of channels, $H, W$ as the height and width of the image. **Channel Filtering:** Calculate channel importance scores using Global Average Pooling (GAP):

$$g_c = \frac{1}{H \times W} \sum_{h=1}^{H} \sum_{w=1}^{W} X_{c,h,w}, \quad \forall c \in \{1, 2, \ldots, C\}$$

Sort channels by importance scores $g_c$ in descending order. Retain the top $N$ most important channels, forming the filtered image:

$$\mathbf{X}^{\text{filtered}} \in \mathbb{R}^{N \times H \times W}$$

**Probabilistic Amplification:** each channel $n$ in $\mathbf{X}^{\text{filtered}}$ Initialize learnable parameters $\mu_n, \sigma_n^2$. each pixel $(h, w)$ in the channel Calculate pixel-wise probability using Gaussian-like distribution:

$$P_{n,h,w} = \exp\left(-\frac{(X_{n,h,w} - \mu_n)^2}{2\sigma_n^2}\right)$$

**Multi-Variate Probabilistic Amplification:** Combine all channel-wise probability matrices with the filtered input:

$$\mathbf{X}^{\text{multi}} = \text{Concat}(\mathbf{X}^{\text{filtered}}, \mathbf{P}_1, \mathbf{P}_2, \ldots, \mathbf{P}_N)$$

Calculate unified multi-variate probability for each pixel:

$$P_{h,w}^{\text{multi}} = \exp\left(-\frac{1}{2} \sum_{n=1}^{N} \frac{(X_{n,h,w} - \mu_{n,h,w})^2}{\sigma_{n,h,w}^2}\right)$$

**Reweighting Block:** Convert the unified probability matrix into a pixel-wise reweighting matrix:

$$W_{h,w} = \frac{\exp(-\lambda P_{h,w}^{\text{multi}})}{\sum_{h=1}^{H} \sum_{w=1}^{W} \exp(-\lambda P_{h,w}^{\text{multi}})}$$

Apply the reweighting matrix to the input image: each channel $c$ in the original image $\mathbf{X}$ each pixel $(h, w)$ Calculate the reweighted pixel value:

$$X_{c,h,w}^{\text{reweighted}} = X_{c,h,w} \times W_{h,w}$$

**Return:** Reweighted image $\mathbf{X}^{\text{reweighted}} \in \mathbb{R}^{C \times H \times W}$

---

# F   TRAINING DETAILS

To address the severe class imbalance in wildfire detection, we adopt a custom F1-based loss function that directly optimizes the harmonic mean of precision and recall. Let $\mathbf{y} \in \mathbb{R}^{B \times C \times H \times W}$ denote the predicted logits and $\mathbf{y}_{\text{true}} \in \mathbb{N}^{B \times H \times W}$ denote the ground truth labels. The targets are converted to one-hot encoded format $\mathbf{y}_{\text{one-hot}} \in \{0, 1\}^{B \times C \times H \times W}$ by mapping each scalar label to a binary vector over $C$ classes. We compute the predicted class probabilities using the softmax function: $\mathbf{p} = \text{softmax}(\mathbf{y})$. For each class $c$, the true positives (TP), false positives (FP), and false negatives (FN) are computed as:

$$\text{TP}_c = \sum(\mathbf{p}_c \cdot \mathbf{y}_{\text{one-hot},c}), \quad \text{FP}_c = \sum(\mathbf{p}_c \cdot (1 - \mathbf{y}_{\text{one-hot},c})), \quad \text{FN}_c = \sum((1 - \mathbf{p}_c) \cdot \mathbf{y}_{\text{one-hot},c}). \quad (41)$$

These are used to calculate precision, recall, and the per-class F1 score:

$$\text{Precision}_c = \frac{\text{TP}_c}{\text{TP}_c + \text{FP}_c + \epsilon}, \quad \text{Recall}_c = \frac{\text{TP}_c}{\text{TP}_c + \text{FN}_c + \epsilon}, \tag{42}$$

$$\text{F1}_c = \frac{2 \cdot \text{Precision}_c \cdot \text{Recall}_c}{\text{Precision}_c + \text{Recall}_c + \epsilon}. \tag{43}$$

The overall F1 loss is defined as the average over all classes:

$$\mathcal{L}_{\text{F1}} = 1 - \frac{1}{C} \sum_{c=1}^{C} \text{F1}_c. \tag{44}$$

This formulation encourages the model to maintain balanced precision and recall, thereby improving sensitivity to sparse but critical fire regions that are often missed by standard loss functions.

All models are trained for 10,000 iterations using the Adam optimizer with an initial learning rate of $1 \times 10^{-3}$, weight decay of $5 \times 10^{-4}$, and a batch size of 8. The learning rate follows a polynomial decay schedule defined as:

$$\text{lr}_t = \text{lr}_0 \cdot \left(1 - \frac{t}{T}\right)^{0.09}, \tag{45}$$

where $\text{lr}_0$ is the base learning rate, $t$ is the current iteration number, and $T = 10000$ is the total number of training iterations. Model performance is validated every 500 iterations, and early stopping is applied if the validation F1 score does not improve for three consecutive evaluations. This training regime ensures stable convergence and allows for consistent comparison across original, compressed, and PAMELA-enhanced model variants.

## G  USE OF LARGE LANGUAGE MODELS (LLMS)

We used ChatGPT (GPT-5, OpenAI) to aid in polishing the writing of this paper. Specifically, it was employed to refine sentence structure, improve clarity, and adjust tone for academic writing. All research ideas, experimental design, implementation, and analysis were entirely conducted by the authors.

