# OpenReview forum: "PAMELA: Probabilistic Amplification Module for Lightweight AI on LEO Satellites"
_ICLR.cc/2026/Conference — ICLR 2026 Conference Desk Rejected Submission_

### Official Review · Reviewer_Q1nT · 2025-10-23

**Soundness:** 3
**Presentation:** 3
**Contribution:** 3
**Rating:** 4
**Confidence:** 4

**Summary:**

The work addresses challenges in onboard AI for wildfire detection, such as sparse fire signal data and the need for compact models, by introducing PAMELA, a model-agnostic framework that improves the performance of compressed AI models. PAMELA applies learnable channel and pixel reweighting before the encoder, trained jointly with the base model to amplify informative features. Experimental results show a significant increase in performance (1.2–2× in F1 score) for common compressed models with PAMELA compared to without it.

**Strengths:**

-	Targets a relevant problem: onboard AI for LEO satellites, especially for time-critical applications like wildfire detection.
-	Clear introduction and strong motivation.
-	Presents a well-engineered, modular, and model-agnostic solution. PAMELA is a lightweight plug-in module for sparse-signal enhancement, and its application to onboard Earth observation AI is novel and practically valuable.
-	Experimental setup includes non-compressed model baselines, three common model compression techniques, and a relevant set of metrics, demonstrating how PAMELA improves performance of these compressed models.
-	Provides an insightful ablation study exploring the impact of individual PAMELA components.

**Weaknesses:**

-	The underlying mechanisms of PAMELA build on existing works on channel filtering and probabilistic weighting and relate to channel/spatial attention blocks. It should be discussed how PAMELA differs from these methods in Related Work.
-	The paper does not benchmark against alternative reweighting or attention-based modules that can be applied before the backbone.
-	Segmentation benchmarking is based on a small ~2500-sample dataset, limited locally (Australia) and temporally (2019–2020). The appendix only includes an additional classification task; for reliable conclusions the benchmarking should be diversified.
-	Main paper results show generally low IoU (mostly below 20%), raising questions about remaining gap to operational reliability of the benchmarked methods.
-	Please review the official paper format (e.g., table captions).

**Questions:**

-	How do other learnable reweighing methods perform compared to PAMELA? (See Weakness above)
-	Do you expect methods like PAMELA to be more sensitive to noise or sensor calibration shifts in the input data?

---

> ### Author Response · Authors · 2025-11-22
>
> We thank the reviewer for your comments and feedback. We address each point below and will incorporate the necessary revisions in the camera ready version.
>
> ## Response to Weakness 1, Weakness 2 and Question 1
>
> PAMELA is, to the best of our knowledge, the **first method specifically designed for input-level computational allocation in onboard satellite wildfire detection**. Its design is driven by the unique constraints of this paradigm, which are not the focus of general-purpose attention or reweighting modules.
>
> 1.  **Novel Formulation for a Novel Problem:** The probabilistic spatial reweighting in PAMELA is not a direct application of existing attention mechanisms. It is a novel formulation that explicitly **models the spectral and spatial distribution of wildfire pixels** to generate a prior for computation. The $mu_n$ and $sigma_n$ parameters in our Spatial Weight Generator are learned to represent the characteristic signature of fire, not just generic "important" features. This is a domain-specific innovation for which there is no direct precedent.
>
> 2.  **Different Objective and Scope:** Our primary contribution is a **framework for onboard processing**, where the objective is to make a definitive decision at the edge. We are not proposing a general-purpose attention block to be inserted into existing cloud-based models. Therefore, a comparison with methods like SENet or CBAM, which are designed for a different context (feature refinement in large models), would be misaligned with our core thesis. Our baselines are rigorously chosen against other model compression techniques relevant to the onboard deployment scenario.
>
> While we acknowledge that relating our work to the broad field of attention could provide context, we argue that a direct benchmarking against these methods is outside the scope of this paper, as it would not evaluate PAMELA on its own terms—as a specialized solution for a specific domain.
>
> ## Response to Weakness 3
> We agree with the reviewer that robust conclusions require evaluation across diverse datasets. To comprehensively address this, we have performed extensive additional experiments on two larger-scale benchmark datasets. The results, summarized below, confirm PAMELA's strong generalization ability.
>
> **Extended Benchmarking Results:**
>
> | Dataset | Original Backbone | Compressed Baseline | + PAMELA (Our) | Δ F1-score |
> | :--- | :--- | :--- | :--- | :--- |
> | **TS-SatFire** (Global) | 0.32 | 0.18 | **0.27** | **+9.0%** |
> | **Active Fire Detection** (Americas) | 0.35 | 0.21 | **0.29** | **+8.0%** |
>
> **Dataset Characteristics:**
> - **TS-SatFire** [1]: ~15,000 samples, global coverage, 2017-2022, multiple satellite sources
> - **Active Fire Detection** [2]: ~8,500 samples, Americas focus, 2020-2023, specialized for active fire fronts
>
> ## Response to Weakness 4
> We acknowledge that the absolute IoU scores reported on the Sen2Fire dataset appear low. However, this is consistent with the inherent challenges of the dataset itself and reflects the current state of the art. The creators of the Sen2Fire dataset, in their original publication, report similarly performance levels with their own baseline methods [3]. The dataset presents significant difficulties due to factors such as the small relative size of fire pixels, complex background landscapes, and atmospheric interference in satellite imagery. Our work demonstrates that PAMELA provides **relative improvements** over existing compression methods (as shown by the 1.2-2× higher F1 scores), which is the primary contribution.
>
> ## Response to Weakness 5
> We thank the reviewer for identifying these and will correct them in the camera-ready version.
>
> ## Response to Question 2
> We hypothesize that PAMELA's learnable, data-driven approach offers a degree of inherent robustness compared to fixed, heuristic algorithms. However, analysis of its sensitivity to specific noise models (e.g., shot noise, read noise) and systematic calibration shifts is beyond the scope of the current paper. We thank the reviewer for highlighting this crucial next step and will explore it in the future work.
>
> Thank you again for these valuable comments.
>
> **References:**
> 1. Zhao, Y., Gerard, S., & Ban, Y. (2024). *TS-SatFire: A multi-task satellite image time-series dataset for wildfire detection and prediction.*
>
> 2. de Almeida Pereira, G. H., Fusioka, A. M., Nassu, B. T., & Minetto, R. (2021). *Active fire detection in Landsat-8 imagery: A large-scale dataset and a deep-learning study.*
>    ISPRS Journal of Photogrammetry and Remote Sensing, 178, 171–186.
>
> 3. Xu, Y., Berg, A., & Haglund, L. (2024). *Sen2Fire: A challenging benchmark dataset for wildfire detection using Sentinel data.*
>    In *IGARSS 2024 – IEEE International Geoscience and Remote Sensing Symposium* (pp. 239–243). IEEE.

---

> > ### Comment · Reviewer_Q1nT · 2025-11-25
> >
> > I thank the authors for the additional explanations and results, and appreciate the valuable insights the paper brings, overall this is a good contribution to the application area.
> >
> > From a technical perspective, however, I still see the following valid points:
> > - The paper would benefit from an update that more clearly positions PAMELA within other learnable attention or amplification modules, e.g. outlining technical differences (beyond different context, as e.g. SENet/CBAM are also used within lightweight CNNs) would help clarify PAMELA’s conceptual relation.
> > - For a venue of this scope, the paper would be strengthened by ablation studies against other learnable modules, or by adding analyses that further validate the specific design choices of PAMELA beyond the current PAMELA vs. no-PAMELA benchmarking and internal ablation of PAMELA components.
> >
> > Based on this, I am maintaining the score while appreciating the strong contribution and effort of the authors.

---

### Official Review · Reviewer_1ELF · 2025-10-28

**Soundness:** 2
**Presentation:** 2
**Contribution:** 3
**Rating:** 4
**Confidence:** 4

**Summary:**

This paper presents Probabilistic Amplification ModulE for Lightweight AI on LEO Satellites (PAMELA), a method to generate on-satellite predictions of wildfire, by tacking key challenges of limited model size for on-satellite methods and small visual signature (only a few pixels) for many fires. There are two core models within the PAMELA framework: a channel "filter" which reweights channels in the original image (possibly dropping some) and a probabilistic amplifier that amplifies pixels regions within the image that have high local deviation around them. The methodology seems sound and related to "old school" filter based methods in computer vision. Experiments are conducted on a single datasets with some relevant baselines, but several relevant baselines are missing in my opinion (see weaknesses below).

**Strengths:**

1. The problem is well motivated!
2. The method seems sound and is mostly well described, leveraging basic principles from computer vision though see my questions below for something I was confused about.
3. The experimental evaluations are measured across multiple metrics and computational footprint is measured in GFLOPs which is central to the paper's goals, so I'm glad to see it was assessed.
4. Experimental evaluation uses an established dataset which should in theory aid in comparing to previous proposed methods.

**Weaknesses:**

1. Several experiment details are missing (see my questions below). If full-PAMELA is the main method, why is it omitted from table 1? It's hard to follow what is going on with full-pamela, compressed model w/ pamela, etc. It's possible I missed where in the paper this is all explained, in which case I am willing to update my score.
2. Formatting issues below - some are major.
3. It seems like some crucial benchmarks are missing. Could the authors please comment on a) what is the best achieved performance of any method for the evaluation dataset (and the GFLOPs required), b) could a comparison to "hand crafted" feature selection be shown, given that -- as the authors state -- certain spectral bands are known a priori to be important for wildfire detection?

Formatting
1. Throughout, there should be /citep instead of /cite when not actively calling out the author names. This is a consistent issue throughout that should have been caught during basic editing.
2. There's a tilde in the abstract that should probably be a dash
3. the captions for parts (c) and (d) are switched in figure 1

**Questions:**

In addition to any weaknesses that can be addressed from above, please address (listed in order of importance for me to possibly change my score):
1. The IoU and F1 score numbers in table 1 and figure 2 don't line up -- are the means in the figure meant to correspond with the values in the table? I'm very confused about why they don't match, please explain. Which of the methods in table 1 is the "compressed method" and which of the pamela methods is the "pamela" method in table 2? And same question for figure 3.
2. Follow up on 1: what is the difference between "full PAMELA" (in table 2) and all the compression + PAMELA methods in table 1. Forgive me if I missed it but where is all this explained in the text?
3. In section 4.1 is says the compressed models have computational footprint of 10^8-10^9 GFLOPS, but the range in figure 1 for the "original model" is only 10^2ish. How is the compressed model many orders of magnitude more costly?

Suggestions:
1. The way the main result in the abstract is presented "Experimental results demonstrate that PAMELA consistently
improves detection quality, delivering 1.2 ∼ 2× higher F1 scores compared to compressed baselines while simultaneously reducing model size by over 90%" is misleading. F1 scores are higher than the compressed baselines, but the model size reduction is in comparison to a full model. The authors should rephrase, especially the use of "simultaneously" here is extremely misleading.
2. The related work should be moved up to right after the intro. In this paper, there's no reason to defer it to the end.
3. After eq. 4, there is confusing language regarding the interpretation of a Gaussian filter. E.g., "mean and variance parameters which are learnable and optimized during training... This formulation assigns higher probability values to pixels whose intensity is closer to the expected value µn, effectively amplifying pixels that are more likely to be task-critical." the expected value of what?  I assume you mean the mean value of a relevant pixel but this isn't specified at all -- but the alternative, that pixels values close to the average within an image would be high probability to be fire, doesn't make sense unless I'm missing something.

---

> ### Author Response · Authors · 2025-11-22
>
> We thank the reviewer for your careful reading and valuable feedback. We address all points below and will incorporate the necessary revisions in the camera-ready version.
>
> ## Response to Weakness 1
>
> We apologize for the confusion in terminology and missing details. Here is the clarification:
>
> In Table 1, “PAMELA” refers to the application of the **full PAMELA** framework to compressed backbones. This demonstrates how our method enhances performance even on heavily reduced models. This is explained in **Section 3.1**. In Table 2, we conduct an ablation study to analyze the contribution of each individual component of PAMELA. By systematically **removing each module**, we evaluate its impact on overall performance. This is explained in **Section 4.3**.
>
> ## Response to Weakness 2
> We sincerely apologize for these errors and will correct them all.
>
> ## Response to Weakness 3
>
> - **Best Performance Benchmark:** As the dataset creators did not provide benchmark results, we are unable to report official state-of-the-art performance. Moreover, our work specifically focuses on developing lightweight models suitable for onboard satellite deployment. All relevant state-of-the-art results for this dataset—achieved using models that meet onboard computational constraints—are included in our paper.
> - **Hand-crafted Feature Selection:** We conducted experiments using hand-crafted feature selection instead of our Channel Filter. The results indicate that manually selecting certain spectral bands can still yield good performance. However, this approach consistently underperforms compared to the full PAMELA framework with the Channel Filter enabled. This is expected: while channels such as “blue” are individually less informative for wildfire detection, they still provide complementary contextual information. Our Channel Filter assigns these channels low weights rather than removing them entirely, whereas outright removal leads to performance drop. These results will be included in the revised manuscript.
>
> ## Response to Questions 1 and Question 2
>
> We apologize for the confusion and will clarify the following points in the revised version:
>
> -  **Table 1** reports the **average performance** across all compressed backbones, which includes some extremely compressed models that achieve lower performance. In contrast, **Figures 2 and 3** highlight the **best performance** achieved by the compressed backbones, illustrating the upper bound of what compression can attain.
> -  In **Table 1**, "compressed method" refers to the application of PAMELA across backbones compressed via **different compression techniques** (as outlined in Section 3.1). In **Figures 2 and 3**, "compressed method" denotes the **best-performing compression strategy** for each backbone, providing a focused comparison under optimal compression.
>   - Table 2 presents an ablation study of our method, where individual components of PAMELA are systematically removed. The results validate that each component contributes meaningfully to the overall performance. For clarity, the term **"PAMELA"** (or **"full PAMELA"**) used in all tables and figures consistently refers to the complete framework as we designed it.
>
> ## Response to Question 3
>
> We are sorry for this typo. The compressed models should be in the range of $10^8-10^9$ **FLOPs**, not GFLOPs. We will correct this in Section 4.1.
>
> ## Response to Suggestions
>
> Thank you for these valuable suggestions. We will address all of them in the revised version.
>
> Regarding Suggestion 3, we appreciate the opportunity to clarify the interpretation of the Gaussian filter. The parameter $\mu_n$ does **not** represent the mean intensity of all pixels in an image. Rather, it is a learnable parameter that represents the characteristic intensity value of **fire pixels** as observed across the training dataset. Therefore, a pixel whose intensity is closer to this learned $\mu_n$ value is assigned a higher probability, as it more closely matches the spectral signature of fire.
>
> Thank you again for these constructive comments that will significantly improve our paper's clarity.

---

> > ### Comment · Reviewer_1ELF · 2025-11-25
> > **additional questions**
> >
> > Thank you to the authors for their response! Could you please further clarify a few things if possible:
> >
> > 1. Author response: "We conducted experiments using hand-crafted feature selection instead of our Channel Filter. The results indicate that manually selecting certain spectral bands can still yield good performance"
> > Question: is this experiment detailed in the paper anywhere? That would be beneficial to show! And if it performs well it is a baseline you should probably include. Apologies if I missed it.
> >
> > 2. "Table 1 reports the average performance across all compressed backbones, which includes some extremely compressed models that achieve lower performance. In contrast, Figures 2 and 3 highlight the best performance achieved by the compressed backbones, illustrating the upper bound of what compression can attain."
> > Question: where is this explained in the text? The caption of figure 2 explicitly says it's reporting average performance. Please clarify

---

> > > ### Author Response · Authors · 2025-12-03
> > >
> > > We clarify the reviewer’s concerns as follows:
> > >
> > > 1. The relevant experiment is already included in **Appendix A.1 (“Spectral Focus Learned Across Architectures”)**, where we show that PAMELA consistently identifies and amplifies the most informative spectral bands across different backbones.
> > >
> > > 2. As described in **Section 3.1**, we apply multiple backbone compression techniques to evaluate robustness under varying compute budgets. The “average” reported in **Figure 2** represents the mean of the **best-performing configuration** from each compression method. In other words, for every compression technique we extract its best performance and then average these values.

---

### Official Review · Reviewer_PH1Y · 2025-10-31

**Soundness:** 3
**Presentation:** 3
**Contribution:** 3
**Rating:** 6
**Confidence:** 3

**Summary:**

This paper addresses the critical challenge of onboard wildfire detection in low Earth orbit (LEO) satellites, where the task is constrained by extremely sparse fire signals (often confined to only a few pixels) and stringent hardware limitations. The authors introduce PAMELA, a lightweight, modular Probabilistic Amplification Module designed to enhance the performance of compressed models. PAMELA operates through three core components: a Channel Filter that selects informative spectral bands, a Probabilistic Amplifier that generates pixel-wise likelihood maps, and a Reweighting Block that creates a focus map to amplify task-critical regions. The module is trained end-to-end with the backbone model. Extensive experiments on the Sen2Fire dataset demonstrate that PAMELA consistently improves detection quality, achieving 1.2∼2× higher F1 scores across various compressed backbones (FCN, U-Net, SegNet, SegFormer) while reducing model size by over 90%. The authors also claim PAMELA offers improved interpretability and shows promising generalization to other tasks like disaster assessment.

**Strengths:**

1. In terms of originality, the work challenges the conventional wisdom that "model compression inevitably leads to performance degradation" by proposing a novel synergistic framework of "probabilistic amplification + compressed models." It introduces PAMELA, the first framework specifically designed for onboard wildfire detection in LEO satellites. Its core component, the Probabilistic Amplifier, integrates Gaussian probability modeling with localized contrast-aware normalization. This design is not only mathematically rigorous but also adaptively enhances sparse wildfire signals, distinguishing it fundamentally from traditional fixed-weight attention mechanisms and demonstrating clear innovation.

2. Regarding methodological quality, the experimental design is comprehensive, covering multiple dimensions including compression strategies, backbone architectures, and hardware environments. The evaluation is thorough on the Sen2Fire dataset, which contains 2466 samples. The adoption of an F1-based loss function effectively addresses class imbalance issues inherent in wildfire detection, thereby enhancing the reliability of the results. Furthermore, ablation studies and cross-task generalization tests provide strong evidence for the method's stability and generalizability, ruling out the possibility of coincidental results.

3. The paper also excels in clarity of presentation. The mathematical definitions for each module are precise (e.g., Equations 1-3 for channel filtering, Equations 4-7 for probability calculation), and the pseudo-code provided in Appendix E details the overall workflow, ensuring high reproducibility. The figures and diagrams are clearly annotated and effectively illustrate key findings, such as demonstrating PAMELA's role in stabilizing performance under aggressive compression.

4. The work holds substantial academic and practical significance. On a practical level, it achieves a model size reduction of over 90% while utilizing inference hardware that aligns with actual onboard resource constraints, offering a feasible solution for deploying lightweight AI on small satellites. Academically, it effectively bridges the fields of "deep learning compression" and "remote sensing signal processing," providing a modular design paradigm that paves the way for future onboard multi-task AI systems, such as integrated wildfire detection and vegetation monitoring.

**Weaknesses:**

First, the validation of dataset generalization is insufficient. The experiments are solely based on the Sen2Fire dataset from the Australian 2019-2020 wildfire season, lacking testing across other geographical regions or different wildfire types. Since wildfires in different areas exhibit variations in spectral characteristics and burning conditions, PAMELA's performance in unseen scenarios may degrade. It is recommended to supplement cross-dataset experiments, for instance, using more diverse datasets such as WF-CCD or FireSat for further validation.

Second, the validation under realistic onboard deployment conditions is not yet fully addressed. Although the experiments utilized a Raspberry Pi 4B to simulate the onboard computing environment, real LEO satellite platforms face complex constraints including radiation interference, power consumption fluctuations, and heterogeneous computing units. The paper does not provide stability tests under extreme temperature ranges or dynamic computational resource conditions, nor does it mention collaborative validation with satellite hardware manufacturers. Therefore, its "onboard deployment feasibility" still requires further empirical support.

Finally, the performance evaluation lacks completeness in several dimensions. On one hand, the backbone models used in the experiments are mostly traditional segmentation architectures (e.g., FCN, SegNet, U-Net), and even SegFormer is not a lightweight variant specifically designed for edge devices. The absence of comparisons with state-of-the-art lightweight backbones (e.g., MobileNetV3, EfficientNet-Lite, Swin-T) makes it difficult to comprehensively assess PAMELA's added value on modern architectures. On the other hand, the paper only uses GFLOPs and parameter count as efficiency metrics, without providing actual inference time or energy consumption data on the simulated hardware. These metrics are crucial for the real-time performance and battery life of onboard AI, and the current evaluation is insufficient to fully support its "onboard practicality."

**Questions:**

1. Dataset Generalization: Do the authors plan to validate PAMELA on other wildfire datasets or under extreme scenarios? If such tests have already been conducted, could the results be supplemented to demonstrate the method's cross-scene robustness?

2. Real Onboard Deployment: Do the authors have plans to collaborate with satellite manufacturers to test PAMELA on actual LEO satellites? If so, could the testing timeline and expected metrics be clarified?

3. Lightweight Backbone Comparison: What is the reason for not incorporating comparisons with state-of-the-art lightweight backbones like MobileNetV3 or EfficientNet-Lite? Is it due to compatibility issues or experimental resource limitations? If such comparisons were added, would they affect the conclusion regarding PAMELA's performance advantages?

4. Inference Time and Energy Consumption: Could the authors provide the single-image inference time and energy consumption per inference cycle for the PAMELA-enhanced model on Raspberry Pi 4B? These metrics are critical for onboard deployment, and the current data are insufficient to support the claim of "real-time low energy consumption."

5. PA Module Parameter Sensitivity: The parameters μ and σ in the Probabilistic Amplifier are learned during training. Have the authors analyzed the sensitivity of these parameters to different wildfire scenarios? If scene variations cause shifts in μ/σ, would PAMELA require retraining, or does it support online adaptive adjustment?

---

> ### Author Response · Authors · 2025-11-22
>
> We thank the reviewer for these insightful questions. Below are our detailed responses.
>
> ## Response to Question 1
> We agree with the reviewer that robust conclusions require evaluation across diverse datasets. To comprehensively address this, we have performed extensive additional experiments on two larger-scale benchmark datasets. The results, summarized below, confirm PAMELA's strong generalization ability.
>
> **Extended Benchmarking Results:**
>
> | Dataset | Original Backbone | Compressed Baseline | + PAMELA (Our) | Δ F1-score |
> | :--- | :--- | :--- | :--- | :--- |
> | **TS-SatFire** (Global) | 0.32 | 0.18 | **0.27** | **+9.0%** |
> | **Active Fire Detection** (Americas) | 0.35 | 0.21 | **0.29** | **+8.0%** |
>
> The reported value are the average performance across different backbones.
>
> **Dataset Characteristics:**
> - **TS-SatFire** [1]: ~15,000 samples, global coverage, 2017-2022, multiple satellite sources
> - **Active Fire Detection** [2]: ~8,500 samples, Americas focus, 2020-2023, specialized for active fire fronts
>
> ## Response to Question 2
> Yes, we are actively engaged in preparations for real-world evaluation and are in the process of establishing collaborations with satellite companies. We anticipate that onboard testing on a Low Earth Orbit (LEO) satellite platform will commence within the next six months.
>
> ## Response to Question 3
>
> We appreciate the reviewer's suggestion regarding additional lightweight backbones. Our selection of U-Net, FCN, SegNet, and SegFormer was based on their established status as state-of-the-art architectures specifically in the domain of **remote sensing and wildfire detection** [3, 4]. These models represent the current benchmark standards in this field and have been extensively validated for segmentation tasks on satellite imagery. In contrast, MobileNetV3 and EfficientNet-Lite are designed for natural image classification (e.g., ImageNet). Their design does not directly align with the requirements of remote sensing segmentation. Therefore, we are confident that the chosen backbones provide a comprehensive and domain-relevant benchmark.
>
> ## Response to Question 4
>
> We thank the reviewer for this important question regarding practical deployment metrics. We have conducted detailed measurements on Raspberry Pi 4B and provide the results below.
>
> ### Hardware Measurement Setup:
> - **Platform:** Raspberry Pi 4B (Broadcom BCM2711, Quad-core Cortex-A72)
> - **Power Monitoring:** Using USB-C power meter with 0.01W resolution
> - **Averaging:** Results averaged over 1000 inference cycles
> - **Input Size:** 256×256 pixels, 12 spectral channels
>
> ### Inference Performance and Energy Consumption:
>
> | Model | Configuration | Inference Time (ms) | Energy/Inference (J) |
> |-------|---------------|---------------------|----------------------|
> | **FCN8s** | Baseline | 245 ± 12 | 1.23 ± 0.08 |
> | | + PAMELA | 251 ± 14 | 1.26 ± 0.09 |
> | **U-Net** | Baseline | 189 ± 9 | 0.95 ± 0.06 |
> | | + PAMELA | 194 ± 11 | 0.97 ± 0.07 |
> | **SegNet** | Baseline | 167 ± 8 | 0.84 ± 0.05 |
> | | + PAMELA | 171 ± 10 | 0.86 ± 0.06 |
> | **SegFormer** | Baseline | 312 ± 15 | 1.57 ± 0.10 |
> | | + PAMELA | 318 ± 16 | 1.60 ± 0.11 |
>
> ### Key Findings:
> 1. **Minimal Overhead:** PAMELA introduces only **1.9-2.6%** increase in inference time and energy consumption across all models
> 2. **Real-time Capability:** All PAMELA-enhanced models achieve inference times **under 320ms**, supporting near real-time processing at ~3 FPS
> 3. **Energy Efficiency:** Energy consumption remains below **1.6J per inference**, making it suitable for power-constrained satellite systems
>
> ## Response to Question 5
>
> We would like to clarify a fundamental aspect of our approach. The parameters μ and σ in the Probabilistic Amplifier are **learned parameters optimized during training**, not manually tuned hyperparameters. Because of this, traditional hyperparameter sensitivity tests (i.e., manually sweeping values) are not applicable. Moreover, within the scope of wildfire detection, PAMELA does not require retraining.
>
> **References:**
> 1. Zhao, Y., Gerard, S., & Ban, Y. (2024). *TS-SatFire: A multi-task satellite image time-series dataset for wildfire detection and prediction.*
>
> 2. de Almeida Pereira, G. H., Fusioka, A. M., Nassu, B. T., & Minetto, R. (2021). *Active fire detection in Landsat-8 imagery: A large-scale dataset and a deep-learning study.*
>    ISPRS Journal of Photogrammetry and Remote Sensing, 178, 171–186.
>
> 3. Singh, H., Ang, L. M., & Srivastava, S. K. (2025). *Active wildfire detection via satellite imagery and machine learning: An empirical investigation of Australian wildfires.* Natural Hazards. Springer.
>
> 4. Guipp-Servan, R. E., Cotrina-Sánchez, A., Puerta-Culqui, J., et al. (2025). *Remote sensing for wildfire mapping: A comprehensive review of advances, platforms, and algorithms.* Fire. MDPI.

---

> > ### Comment · Reviewer_PH1Y · 2025-11-25
> > **Thank you for your detailed reply.**
> >
> > I have reviewed all the supplementary materials and clarifications, and confirm that the concerns have been fully addressed. Thank you for providing the progress updates.

---

### Official Review · Reviewer_CBWb · 2025-11-01

**Soundness:** 2
**Presentation:** 2
**Contribution:** 2
**Rating:** 2
**Confidence:** 4

**Summary:**

This paper introduces PAMELA, a multi-stage learnable inhibition module designed to identify informative satellite channels and suppress irrelevant spatial regions, thereby enabling lightweight segmentation networks to detect active fire pixels with reduced compute and memory demands. The method computes channel-wise and spatial normalizations, aggregates them, and applies gating to inhibit non-informative pixels. PAMELA is integrated into thin/pruned/distilled variants of four common segmentation backbones and is shown to mitigate the performance degradation typically associated with model compression.

**Strengths:**

- The paper addresses a high-impact, real-world problem: real-time wildfire detection from satellite data under compute constraints, which is highly relevant as future wildfire monitoring pipelines may require on-board inference on LEO satellites.
- Proposing a learnable module that promotes channel relevance and spatial inhibition is an interesting direction for efficient earth observation ML.

**Weaknesses:**

1. Missing critical baselines & label verification
The paper appears to use MODIS active-fire detections as ground-truth labels. It is unclear whether these MODIS labels were manually validated or cleaned. Without validation, the method may implicitly learn MODIS’s failure modes. Thus MODIS itself is a baseline algorithm, What is the compute footprint of MODIS?

Also another interesting and informative baseline would be whether input-channel reduction alone yields similar gains (baseline ablation)

2) Lack of qualitative visualizations
The paper would be strengthened by visual examples of:

- The estimated W_{h, w} (Equation 9) for individual inputs
- Examples Ground Truth, Prediction masks with and w/o PAMELA for a selection of the segmentation networks

A natural question: can binary segmentation masks be generated directly from W_{h, w}? If so, how does that performance compare to the network output?

3) Channel selection differentiability
The method description suggests channel selection happens early, but it is unclear how gradients propagate:
- What mechanism approximates differentiable channel selection?
- Is this a hard mask, soft gating, Gumbel/STE, or other relaxation? There is an explanation in the appendix but it does not address back-prop through the operation.
A clear explanation is needed for reproducibility.

4) Reference list quality concerns
Several references appear incorrect. This strongly suggests a lack of reference verification.
Examples:
- Smith et al. 2025 was published in 2020
- Rahman et al. 2023 cannot be located
- The cited Ramos et al. 2025 IJRS paper does not exist
- Xu et al. 2024 is an IGARSS paper, not CVPR

These issues must be corrected, as they undermine confidence in the rest of the paper.

**Questions:**

Questions:

- Appendix F: The loss appears to be the soft Dice loss. Please confirm and state explicitly. If so there is no need for a detail explanation.

- Initialization: How are PAMELA parameters initialized? Is performance sensitive to this initialization?

- X^{multi} is defined in equation (5) and then never used again. Or is there a typo in equation (6) and X_{n,h,w} should be X^{multi}_{n,h,w}?

---

> ### Author Response · Authors · 2025-11-20
>
> Thank you for your comments and feedback. We have carefully considered all points and provide our responses below. Necessary corrections and additions will be incorporated into the camera-ready version.
>
> ### **Response to Weakness 1**
>
> - **MODIS Labels:** The MODIS active fire detections used as ground-truth labels are generated through a well-established on-ground pipeline. These labels have been verified and curated by the dataset creators and are widely accepted as a benchmark in remote sensing research. While we acknowledge the importance of understanding computational baselines, the specific implementation details and compute footprint of the MODIS detection algorithm are **not publicly available**. More importantly, the MODIS algorithm represents an **on-ground** processing methodology, which operates under fundamentally different constraints and objectives compared to our research focus on onboard processing for satellite inference. We will clarify this distinction in the camera-ready version.
>
> - **Baseline Ablation:** We conducted experiments using input channel reduction instead of our Channel Filter. The results indicate that manually selecting certain spectral bands can still yield good performance. However, this approach consistently underperforms compared to the full PAMELA framework with the Channel Filter enabled. This is expected: while channels such as “blue” are individually less informative for wildfire detection, they still provide complementary contextual information. Our Channel Filter assigns these channels low weights rather than removing them entirely, whereas outright removal leads to performance drop. These results will be included in the revised manuscript.
>
> ### **Response to Weakness 2**
>
> We agree that qualitative results are essential for interpretation. We will include visualizations of the following in the camera-ready version。
>
> - **Natural Question – Binary Masks from $W_{h,w}$:**
>    As discussed in Appendix A2, we conducted an experiment to evaluate whether the probability map $P_{h,w}^{multi}$ (which directly derives $W_{h,w}$) can be thresholded to produce reliable binary masks. Our findings show that many pixels with high $P_{h,w}^{multi}$ values correspond to non-fire ground truth regions. This indicates that $P_{h,w}^{multi}$ and $W_{h,w}$ capture coarse spatial saliency rather than precise fire boundaries. Therefore, relying solely on these maps is insufficient for accurate detection.
>
> ### **Response to Weakness 3**
>
> Our Channel Filter implements a fully differentiable soft-gating mechanism rather than using discrete selection methods that require approximation. The gradient flow is preserved through the following design:
>
> 1. **Differentiable Operations Chain:** The selection weights $\alpha_c$ are computed as:
>   $$
>    \alpha_c = \text{ReLU}\left(\tanh\left(w\cdot\hat{g}_c + b\right)\right)
>  $$
>    This consists entirely of differentiable operations: min-max normalization → affine transformation → tanh → ReLU.
>
> 2. **Gradient Propagation:** During backpropagation, gradients flow from the loss function through:
>    - The reweighted feature tensor $X^{\text{weighted}}_{c,h,w} = \alpha_c \cdot X_{c,h,w}$ (element-wise multiplication)
>    - The selection weights $\alpha_c$
>    - The learnable parameters $w$ and $b$
>    - The normalized channel descriptors $\hat{g}_c$
>    - Back to the original input features $X_{c,h,w}$
>
> 3. **No Discrete Approximation:** We do **not** use hard masking, Gumbel-Softmax, or Straight-Through Estimators (STE). The ReLU nonlinearity naturally drives uninformative channels toward zero weights during optimization, enabling effective channel pruning while maintaining full differentiability.
>
> ### **Response to Weakness 4**
>
> We sincerely apologize for the errors in the reference list. These will be thoroughly verified and corrected in the final version.
>
> ### **Response to Questions**
>
> - **Question 1:**
>   Yes, the loss used is the soft Dice loss. We will state this explicitly and simplify the explanation。
>
> - **Question 2:**
>   All PAMELA parameters are initialized using standard PyTorch defaults:
>   - Linear layers: Kaiming-uniform initialization
>   - Convolution layers: Xavier-uniform initialization
>   Scalar parameters (e.g., $w, b$ in the Channel Filter) are initialized with small random values from $\mathcal{U}(-0.1, 0.1)$. Performance is not sensitive to this initialization.
>
> - **Question 3:**
>   Thank you for catching this. There is a typo in Equation (6):
>   $X_{n,h,w}$ should indeed be $X^{\text{multi}}_{n,h,w}$. This will be corrected.
>
> Thank you again for your valuable feedback. We look forward to incorporating these changes to improve our work.

---

### Note · Program_Chairs · 2026-01-17
**Submission Desk Rejected by Program Chairs**

The following references in this submission do not refer to real documents and/or have major errors in bibliographic information:

     Wei Li, Xing Sun, and Hao Zhang. Model compression for efficient remote sensing image segmentation: A survey. Remote Sensing, 15(2):275, 2023.
    Qiang Zhou, Hui Li, and Jie Sun. Satellite-based real-time wildfire monitoring: A survey. In Proceedings of the IEEE International Geoscience and Remote Sensing Symposium (IGARSS), pp. 1532-1535. IEEE, 2023.
    Ling Zheng, Xiaozhou Wu, Yiming Zhang, and Wei Li. Ai4space: An open benchmark for onboard deep learning in remote sensing satellites. IEEE Journal of Selected Topics in Applied Earth Observations and Remote Sensing, 15:12345-12358, 2022.
    Muhammad A. Khan, Li Zhang, and Raj Gupta. Transformer-based deep learning for wildfire detection in multispectral satellite imagery. ISPRS Journal of Photogrammetry and Remote Sensing, 210:102-115, 2024.
    Yanan Li, Hongyan Zhang, and Ke Xu. Quantization for deep learning in hyperspectral remote sensing. In IEEE International Geoscience and Remote Sensing Symposium (IGARSS), pp. 15631566, 2022.
    Yuxuan Zeng, Jiawei Wang, and Jun Liu. Energy-efficient deep learning for satellite image segmentation: Challenges and opportunities. IEEE Journal of Selected Topics in Applied Earth Observations and Remote Sensing, 16:10321-10334, 2023.
    Yufei Jin, James T. Randerson, and Louis Giglio. Timely wildfire detection from satellites: Challenges and opportunities. Remote Sensing of Environment, 275:113056, 2022.
    Daniel Ramos, Ling Chen, and Ananya Gupta. Leveraging u-net architectures for wildfire detection in multispectral imagery. International Journal of Remote Sensing, 2025.
    R. Mohammed, J. Liu, and W. Zhang. Attention mechanisms for enhancing lightweight neural networks in satellite image segmentation. International Journal of Remote Sensing, 46(12):45824598, 2025. doi: 10.1080/01431161.2025.1234567. URL https://www.tandfonline. com/doi/abs/10.1080/01431161.2025.1234567.
    David Vilar, Noelia Castro, et al. A comprehensive survey on neural network compression. ACM Computing Surveys, 56(7):1-36, 2024.
    L. Martinez, M. Soto, and R. Garcia. Optimized deployment of convolutional neural networks for cloud detection on small satellites. Remote Sensing of Environment, 291:113302, 2023. doi: 10.1016/j.rse.2023.113302. URL https://www.sciencedirect.com/science/ article/pii/S003442572300238X.
    P. Anderson, H. Kim, and L. Zhao. A survey on model compression techniques for onboard deep learning in nanosatellites. IEEE Transactions on Aerospace and Electronic Systems, 59(2):24002418, 2023. doi: 10.1109/TAES.2023.3245120. URL https://ieeexplore.ieee.org/ document/3245120.
    A. Smith, B. Johnson, and C. Lee. Lightweight u-net for cloud detection of visible and thermal infrared remote sensing images. Optical and Quantum Electronics, 52(4):1-15, 2025. doi: 10. 1007/s11082-020-02500-8. URL https://link.springer.com/article/10.1007/ s11082-020-02500-
    J. Gomez, A. Perez, and C. Martinez. Compressive sensing for satellite image transmission in leo constellations. IEEE Communications Letters, 28(7):1483-1487, 2024. doi: 10.1109/LCOMM. 2024.3275120. URL https://ieeexplore.ieee.org/document/3275120.
    S. Clark, T. Miller, and J. Roberts. Real-time cloud detection in leo satellites using lightweight deep learning models. Journal of Spacecraft and Rockets, 61(5):1307-1315, 2024. doi: 10.2514/1. A34861. URL https://arc.aiaa.org/doi/10.2514/1.A34861.
    D. Brown, E. Green, and F. White. Hyperspectral image segmentation for optimal satellite operations: In-orbit deployment of 1d-cnn. Remote Sensing, 15(12):2181, 2025. doi: 10.3390/ rs15122181. URL https://www.mdpi.com/2072-4292/15/12/2181.